# Associations between dimensions of behaviour, personality traits, and mental-health during the COVID-19 pandemic in the United Kingdom

Adam Hampshire [1 ✉], Peter J. Hellyer [1,2], Eyal Soreq[1], Mitul A. Mehta [2], Konstantinos Ioannidis [3,4], William Trender [1], Jon E. Grant[5] & Samuel R. Chamberlain [6,7]

The COVID-19 pandemic (including lockdown) is likely to have had profound but diverse implications for mental health and well-being, yet little is known about individual experiences of the pandemic (positive and negative) and how this relates to mental health and well-being, as well as other important contextual variables. Here, we analyse data sampled in a large-scale manner from 379,875 people in the United Kingdom (UK) during 2020 to identify population variables associated with mood and mental health during the COVID-19 pandemic, and to investigate self-perceived pandemic impact in relation to those variables. We report that while there are relatively small population-level differences in mood assessment scores pre- to peak-UK lockdown, the size of the differences is larger for people from specific groups, e.g. older adults and people with lower incomes. Multiple dimensions underlie peoples' perceptions, both positive and negative, of the pandemic's impact on daily life. These dimensions explain variance in mental health and can be statistically predicted from age, demographics, home and work circumstances, pre-existing conditions, maladaptive technology use and personality traits (e.g., compulsivity). We conclude that a holistic view, incorporating the broad range of relevant population factors, can better characterise people whose mental health is most at risk during the COVID-19 pandemic.

[1] Imperial College London, London, UK. [2] King's College London, London, UK. [3] Cambridgeshire and Peterborough NHS Foundation Trust, Cambridge, UK. [4] Department of Psychiatry, University of Cambridge, Cambridge, UK. [5] Department of Psychiatry, University of Chicago, Chicago, IL, USA. [6] Department of Psychiatry, University of Southampton, Southampton, UK. [7] Southern Health NHS Foundation Trust, Southampton, UK. ✉email: a.hampshire@imperial.ac.uk

The coronavirus disease 2019 (COVID-19) pandemic has brought about unprecedented change in peoples' lives due to the direct and indirect consequences of illness, physical distancing and socio-economic restructuring. These changes have likely affected mood and mental health in widespread, profound but idiosyncratic ways[1]. Expert groups have posited that the impact of the pandemic on mental health is modulated by a variety of factors, including (i) aspects of demographics such as age or ethnicity, (ii) social networks, (iii) financial and occupational circumstances (iv), being shielded or having carer responsibilities, (v) pre-existing mental health symptoms, (vi) maladaptive online technology use, (vii) personality traits and (viii) tendency towards compulsive behaviours[2–7].

There have been urgent calls to study these relationships as they are critical to inform policy and healthcare decisions and to guide researchers and clinicians. However, pre-existing knowledge about how pandemics affect mental health is limited. Previous work on the relationship between COVID-19 and mental health has focussed on relatively narrow aspects of mental health, failing to consider the diverse psycho-socio-economic variables that are likely to modulate impact, and not taking into account self-perceived impact of the pandemic (positive and negative)[8–12]. Consequently, it remains unclear which segments of society have been most affected or whether expert opinions align with population perspectives. Additionally, extant studies predominantly have used promotional materials that explicitly mention COVID-19, and target established cohorts, yielding high likelihood of recruitment bias from already non-representative subpopulations[13].

Addressing these issues presents a major methodological challenge. There are likely to be multivariate relationships between the ways people have been affected by the pandemic and their psycho-socio-economic profiles. Many of the relevant variables, e.g. personality or psychiatric traits and technology use, covary[14,15], and it is not clear what the major dimensions of the impact are. In such a context, identifying, disentangling and mapping the key variables can only be achieved in a data-driven multivariate manner, necessitating the analysis of large-scale population data.

To address this challenge, we applied a combination of multivariate and machine learning methods to analyse our large-scale data set, comprising a survey of mental health and wellbeing variables completed by 379,875 people, ~90% resident within the UK (see Supplementary Fig. 7) and the remainder from around the globe, since late December 2019 and concentrated around January and May 2020. This broader database includes responses to a questionnaire instrument comprehensively probing self-perceived pandemic impact in 79,779 out of the 112,046 respondents in May–June 2020.

Our analyses comprised several distinct steps designed to address some of the most pressing questions at this time (Fig. 1 top). Specifically, we first confirmed whether there were differences in population distributions of standard depression, anxiety, tiredness and sleep measures between January 2020 (immediately prior to the COVID-19 outbreak reaching the UK), and May 2020 (during peak lockdown in the UK). We examined how the scale of these differences covaried with relevant population factors such as age, education, occupational and socioeconomic status. Next, we quantified prevalent views and estimated in a data-driven manner the dimensionality of self-perceived impact of the pandemic on daily life during lockdown in May 2020. We tested the degree to which there were both positive and negative aspects of self-perceived impact and whether these had significant explanatory value in terms of the standard mental health assessment measures. Finally, we used multivariate analyses to quantify the relative and collective importance of population factors in predicting the idiosyncratic ways that people's daily lives have been perceptibly affected by the pandemic.

## Results

**Respondents and pre-analysis filtering.** The study website, where participants completed cognitive tests and a detailed questionnaire (Supplementary Methods), was made available online from December 26, 2019 at https://gbit.cognitron.co.uk. Articles describing the study were placed on the BBC2 Horizon page, BBC Homepage, BBC News Homepage and circulated on mobile news meta-apps from January 1, 2020 in the UK. A second promotional drive, placing articles in the same prominent locations, was launched on May 2, 2020, aligned with a BBC2 Horizon documentary focussed on concurrently collected cognitive data, analysis of which is outside the scope of this article. This produced large peaks in data collection (Fig. 1 bottom) around the two launches (UK pre-pandemic, 25 December–31 January, $N = 243,875$; mid-pandemic, 2 May–30 June, $N = 125,177$), with smaller but not insubstantial quantities of data in the intervening (early-pandemic, 1 February–1 May, $N = 10,003$) and pre-launch ($N = 820$) periods.

Participants aged <16 years were excluded prior to analysis because they were presented with an abbreviated questionnaire for ethical reasons. Also, as a quality control people who completed the questionnaire in <4 min were excluded; this threshold was decided by the study team prior to database lock based on the approximate minimum time taken to complete the survey while reading the questions. This resulted in 233,268 pre-pandemic, 9049 early-pandemic and 112,046 mid-pandemic participants in the analyses presented here.

A critical question was whether the cross-sectional population samples were well matched between these three epochs. Plotting the sociodemographic distributions (Supplementary Figs. 1–8 and Supplementary Table 1) confirmed generally close correspondences in sex ('what is your sex?' with response options 'male, female or other'), handedness, first language, UK residence, ethnicity, education level, earnings and occupational status across all three epochs. However, the smaller early-pandemic sample, when the study website was not being actively promoted, had a marked skew towards younger age; therefore, we did not analyse this period further herein and rather focussed on the two very closely matched, larger $N$ epochs.

The two epochs of interest captured a broad and inclusive cross-section of the population, with >50 participants per age year per subset up to and including 85. Participants above this age were combined into an 86+ category. Overall, ~14.3% of participants identified as from minority ethnic groups. This inclusivity compares favourably to established UK bioresource cohorts, which often under-represent such groups (e.g. 5.4% ethnic minority groups in the UK Biobank)[16]. There were 45.8% females, 53.3% males and 0.9% indicating other.

**Differences from pre- to mid-pandemic in mean population mood scores.** Mental health self-assessment items were scored on a frequency scale in response to the question of how often the participant had experienced the specified symptom (e.g. feeling down or depressed) over the past weeks. Raw percentage of responses for the pre- and mid-pandemic epochs are reported in (Fig. 2). On visual inspection, the largest difference was for anxiety, with the number of people reporting feeling anxious or on edge 'several times a week' increasing from 24 to 33% and the number of people reporting 'never' decreasing from 18 to 8%. Other aspects of mental health measures showed modest reductions, whereas sleep showed a small increase.

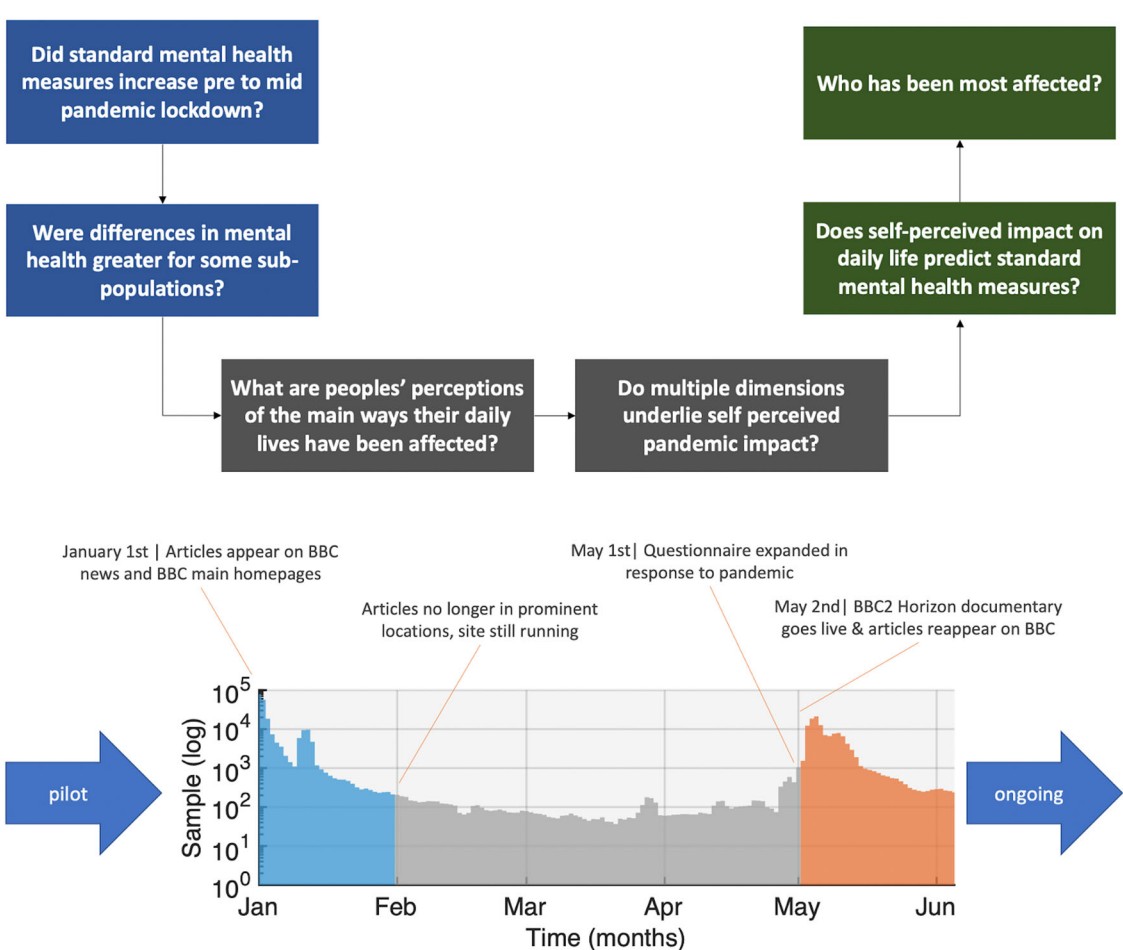

**Fig. 1 Study design.** Top. We first examine whether standard mental health self-assessment measures differed between the period of time just prior to the pandemic hitting the UK and mid-lockdown, and whether those differences covaried with population factors such as age, education and occupational status. We then analyse peoples' self-perceptions of the impact of the pandemic, determining the internal structure and dimensionality of self-perception, and the covariance with population and contextual and lifestyle factors. Bottom. Timecourse of data collection at the time of submission (note, Y axis is a log scale). Large spikes in data collection occurred in January prior to the pandemic hitting the UK and in May at the peak of the first UK-wide lockdown. These pre- and mid-pandemic epochs form the focus of analyses reported in this article. Source data are provided as a Source data file.

Responses were transformed into a numeric scale from 0 (never) to 6 (>hourly) in order to enable statistical analysis of differences in mean mood assessment scores across the mid vs pre-pandemic epochs while accounting for population factors. More specifically, age, sex, handedness, education level, first language, country of residence, occupational status and income were factored out using a general linear model (GLM) from the entire data set for responses reporting the frequencies over the past month of anxiety, depression, problems concentrating, insomnia, hours slept and tiredness. The residual scores were grouped by pre- vs mid-pandemic epoch and compared using $F$ tests. When analysing large-scale data, negligible scaled effects can have significant statistical values, necessitating a focus on effect size to gauge relevance. Here we report mean differences in standard deviation (SD) units, i.e., equivalent to Cohen's $D$, and conform to Sawilowsky's revised[17] criteria, whereby effects of 0.1 SD = very small, 0.2 SD = small, 0.5 SD = medium, 0.8 SD = large, 1.2 SD = very large and 2.0 SD = huge.

The mean differences in mental health measures were generally in the small to very small effect size range. The most pronounced difference was anxiety score, which was greater in the mid-pandemic epoch (0.28 SD, $F(1,3.4425e + 05) = 6090.60$, $p <$ 0.0001). Mean depression score was lower by a very small margin ($-0.08$ SD, $F(1,3.4425e + 05) = 537.75$, $p < 0.0001$) and problems concentrating were greater by a small margin (0.08 SD, $F(1,3.4425e + 05) = 507.88$, $p < 0.0001$). Problems getting to sleep or staying asleep showed a negligible-scaled difference (0.003 SD, $F(1,3.4425e + 05) = 0.628$, $p = 0.4280$); however, reported hours slept per night increased (0.14 SD, $F(1,3.4425e + 05) = 2035.70$, $p < 0.0001$) and a decrease in tiredness was evident ($-0.15$ SD, F $(1,3.4425e + 05) = 1797.30$, $p < 0.0001$). Analysis of day-by-day data during the 31 days after January and the 31 days after May promotion launches (where daily number of participants was high) confirmed that the observed mean differences in mental health measures were consistent throughout those months (Supplementary Fig. 9).

**Population factors covary with differences in mental health scores pre- and mid-pandemic.** We next examined whether differences in mental health pre- to mid-pandemic lockdown were greater for select sociodemographic subpopulations. GLMs were conducted on each of the six item scores. These included interactions of epoch (pre vs mid) with age, sex, handedness, ethnicity, employment status, first language, country of residence,

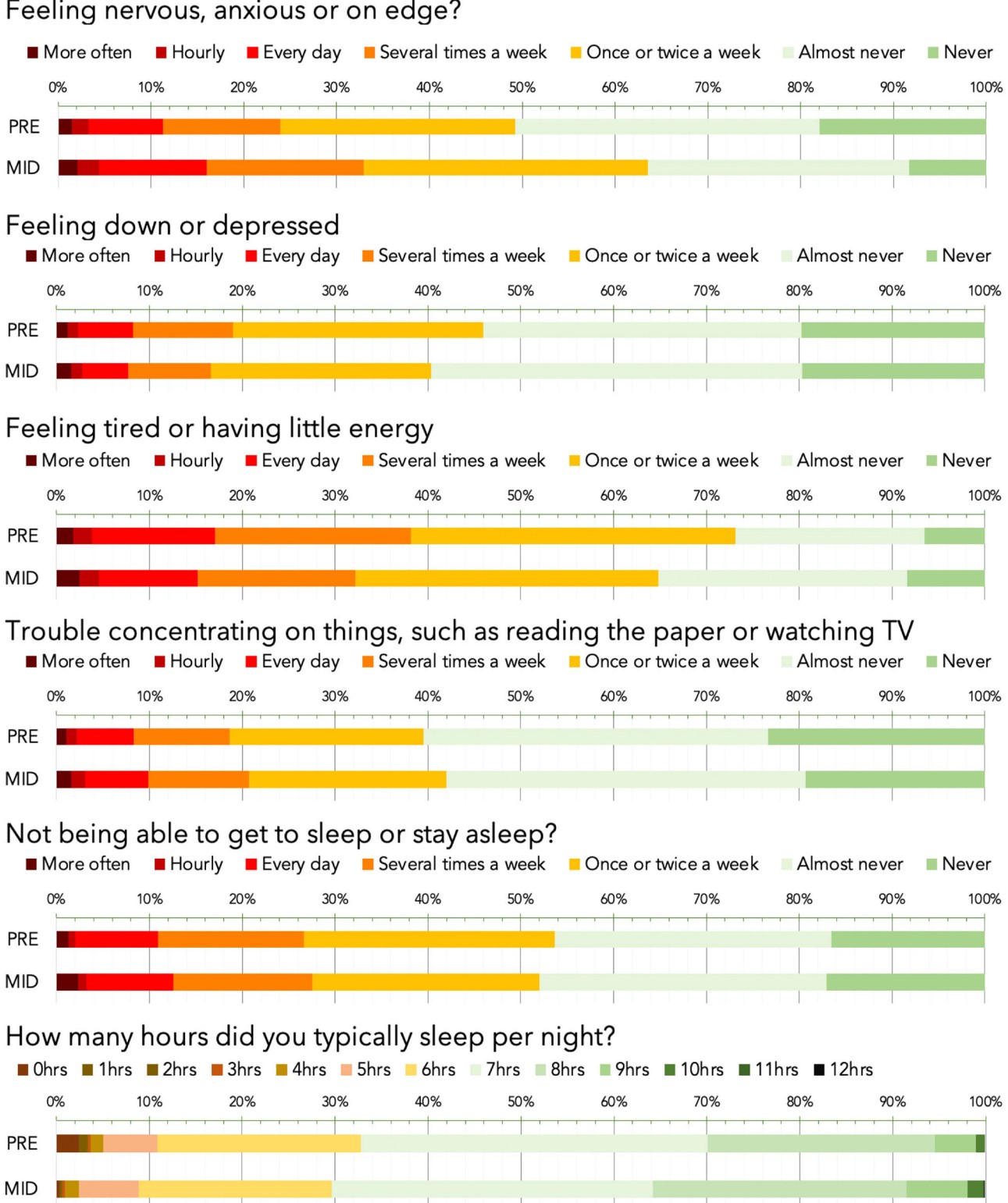

**Fig. 2 Visual comparison of mental health and sleep measures during the pre-UK pandemic and mid-UK lockdown epochs.** Symptoms of mood and sleep problems, as quantified using standard mood self-assessment items, in the population for pre-pandemic (pre) and mid-pandemic (mid). Bars in the top five plots represent the percentage of respondents reporting symptoms at different frequencies over recent weeks. The lower plot reports percentage for typical number of hours slept. On average, anxiety levels were higher and people were sleeping more, whereas other mood symptoms were marginally lower. Source data are provided as a Source data file.

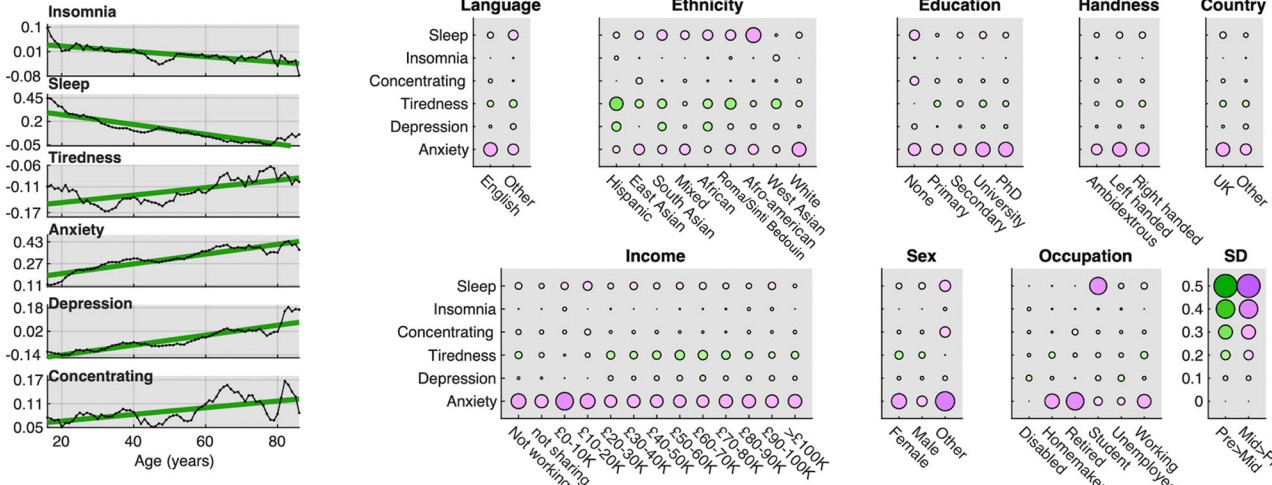

**Fig. 3 Modulation of differences in national mental health scores by population variables.** Left. Differences in mean mental health scores mid-pandemic minus pre-pandemic as a function of age plotted with five-data point smooth and best fit line in green. Y axis in SD units. Older adults showed a greater increase in anxiety. Younger adults showed increased sleep. Younger adults showed decreased depression, whereas older adults showed increased depression. Right. Subpopulation scale and valence of mood-score change. Circles size represents SD units and colour direction of difference (green: pre > mid, pink: mid > pre). Substantial differences were evident as a function of sociodemographic subgroups, with heightened anxiety particularly in retired people, workers, people defining themselves as homemakers, people with lower incomes, and for sex Other vs Female vs Male. Students reported sleeping longer mid-pandemic lockdown. Source data are provided as a Source data file.

education level and earnings, while partialling out the main effects. There were numerous statistically significant interactions (Supplementary Tables 2–16 and Fig. 3) spanning from very small to large effect sizes. Most notable were the relationships with age. There was a simple linear relationship between difference in anxiety score mid- vs pre-pandemic and age (+0.40 SD for 60–80-year-olds, +0.10 SD for teenagers). This relationship also was evident for depression scores, with older adults also showing a small increase (+0.14 SD for 80+), whereas teenagers and young adults showed a small decrease (−0.12 SD). It should be noted that, in both cases, anxiety and depression scores already were higher by a large margin for teenagers and young adults than older adults pre-pandemic lockdown (pre-phase 16–26 minus 76–86 years: anxiety 0.89 SD, depression 0.82 SD). There was a greater difference in hours slept for younger adults (+0.49 h at 16) vs older adults (~0 SD difference at 60+). The pre- to mid-pandemic differences in anxiety score was greatest for retired people (0.38 SD), followed by homemakers (0.31 SD) and workers (0.29 SD). The difference in anxiety scores was greatest for those identifying their sex as other (0.39 SD), followed by females (+0.33 SD) then males (+0.22 SD). Difference in hours slept was greatest for students (+0.37 SD), contrasting with retired people (−0.01 SD and homemakers (+0.02 SD) and disabled people (~0 SD). The relationship between differences in mood scores across epochs and earnings, handedness, first language and country of residence were generally of very small to negligible scale.

**Self-perceived pandemic impact at the level of individual questions.** The results of the cross-epoch analyses of mental health assessment measures accorded with the view that population factors such as age, sex and occupational status may moderate the impact of the pandemic on mood and mental health. Complementing this, in May 2020, the online questionnaire was extended with a bespoke PanDemic General Impact Scale (PD-GIS), designed to probe self-perceived current and longer-term impact of the pandemic on day-to-day life by measuring levels of agreement with positive and negative statements. This optional section of the questionnaire was collected in the

same session. Those who did vs did not complete the PD-GIS had very similar mental health scores (i.e. of very small effect size difference), indicating that sampling bias was negligible (Supplementary Results 1).

We sought to determine which individual items of the PD-GIS people were most likely to agree with (Fig. 4). Unexpectedly, some of the strongest agreement was with positive statements. More than 70% of respondents indicated that they were spending less and saving more, ~50% had more time as were commuting less, >45% more relaxed than before, >65% enjoying the simpler things in life, ~55% had more time to read for pleasure, >80% were more in touch with loved ones they previously had trouble finding time for using apps, >75% noted more pleasant environment with ~70% reporting more wildlife and >65% a greater sense of community. There was strong agreement that technology, science and healthcare would advance more rapidly (>50%) and that things would change but not necessarily for the worse (>60%) but strong disagreement that economic impact would be temporary (60%).

Among the strongest agreement with negative statements was concern for health of loved ones (>85%), which was higher than concern about one's own health (>50%), loss of leisure and health activities (>75%) and disconnectedness (60%) and loss of daily structure (>45%). The strongest disagreement was predominantly with negative statements, e.g. loss of employment (~65%), loss of productivity (~70%), reduced attention to personal hygiene (~65%), loss of access to basics (~70%) and items pertaining to increased conflict at home (~65%). Approximately 30% of people reported drinking more alcohol. Approximately 40% worried about having a less healthy lifestyle.

These measures of overall agreement or disagreement were contextualised by substantial population variability. Therefore, peoples' self-perceptions of the impact of the pandemic on their daily lives were highly variable and included an unforeseen degree of positivity.

**Self-perceived impact of the pandemic at the level of latent variables.** We examined whether the profile of population variability in self-perceived pandemic impact was best captured

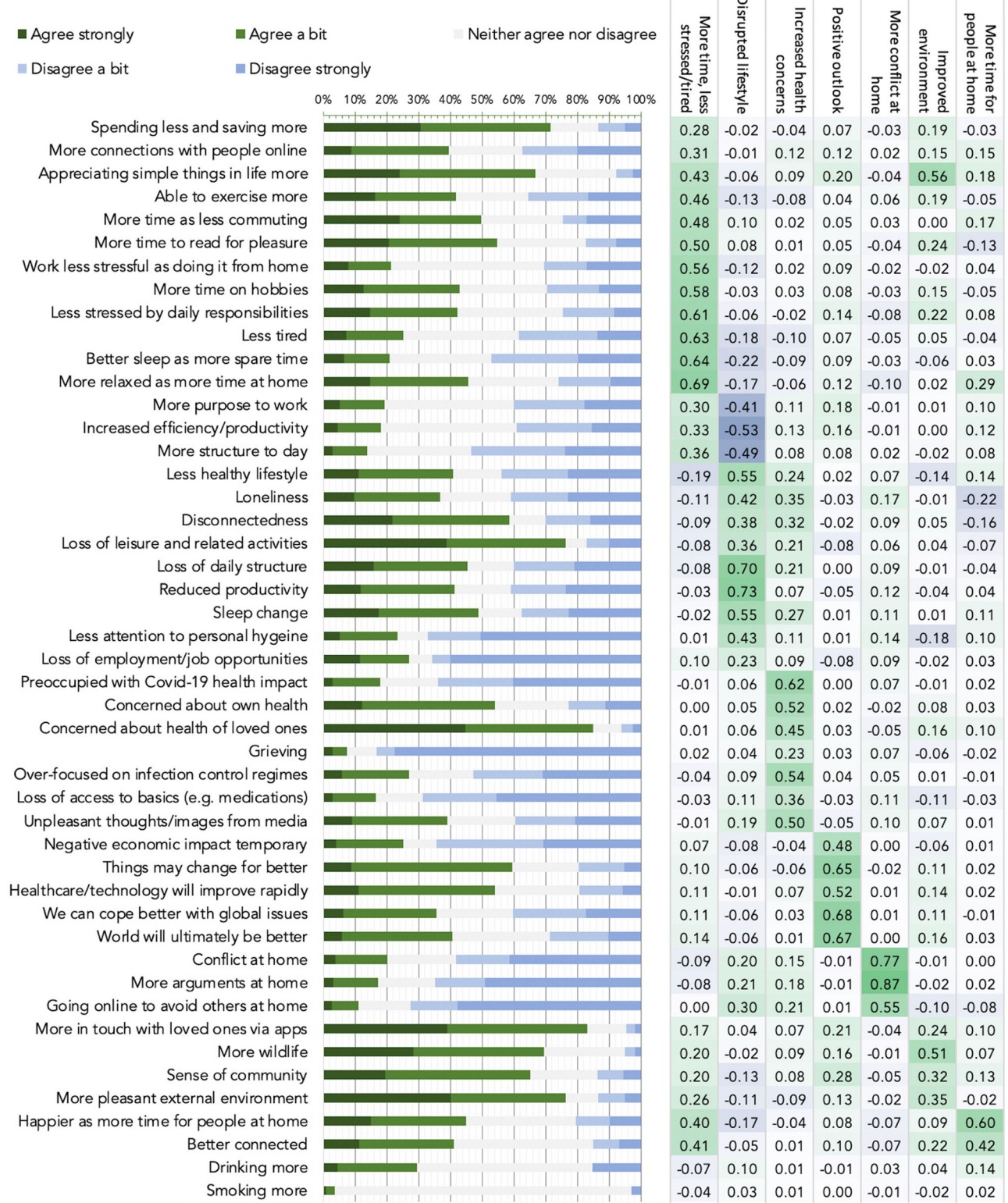

**Fig. 4 Individual item responses and principal component analysis for the PD-GIS at peak UK lockdown.** Left. Strength of agreement with statements about self-perceived pandemic impact—bars are proportion responses. Green = agree, blue = disagree. Abbreviated items are reported in full in Supplementary Methods. Right. Loadings from PCA, which identified 7 components underlying PD-GIS responses when applying permutation testing (Supplementary Fig. 10). These were labelled 1: more time and less stress and tiredness. 2: disrupted lifestyle. 3: increased health concern. 4: positive outlook. 5: more conflict at home. 6: improved environment. 7: more time for people at home. Source data are provided as a Source data file.

**Mental health assessment**

| Pandemic General Impact Scale | Nervous, anxious or on edge | Can't stop worrying | Worry about different things | Trouble relaxing | Restless, can't sit still | Annoyed/irritable | Afraid something awful might happen | Little pleasure doing things | Down or depressed | Tired/ little energy | Trouble concentrating | Can't get to sleep or stay asleep | Hrs slept per night | | Canonical r | p |
|---|---|---|---|---|---|---|---|---|---|---|---|---|---|---|---|---|
| More time, less stressed/tired | -0.11 | -0.09 | -0.09 | -0.15 | -0.07 | -0.10 | -0.06 | -0.10 | -0.12 | -0.17 | -0.07 | -0.18 | 0.23 | | 0.56 | p<0.0001 |
| Disrupted lifestyle | 0.25 | 0.23 | 0.23 | 0.24 | 0.20 | 0.21 | 0.18 | 0.32 | 0.32 | 0.29 | 0.29 | 0.23 | 0.02 | | 0.33 | p<0.0001 |
| Health concerns | 0.29 | 0.30 | 0.30 | 0.27 | 0.18 | 0.14 | 0.36 | 0.17 | 0.23 | 0.20 | 0.19 | 0.20 | -0.12 | | 0.26 | p<0.0001 |
| Positive outlook | -0.05 | -0.04 | -0.03 | -0.03 | 0.00 | -0.05 | -0.04 | -0.04 | -0.05 | 0.00 | -0.02 | -0.01 | 0.01 | | 0.23 | p<0.0001 |
| Conflict at home | 0.13 | 0.12 | 0.12 | 0.12 | 0.11 | 0.31 | 0.10 | 0.15 | 0.16 | 0.11 | 0.12 | 0.05 | 0.02 | | 0.15 | p<0.0001 |
| Improved environment | 0.03 | 0.01 | 0.02 | 0.02 | -0.01 | 0.01 | 0.00 | -0.11 | -0.05 | -0.05 | -0.05 | 0.02 | -0.03 | | 0.12 | p<0.0001 |
| More time for people at home | 0.06 | 0.04 | 0.06 | 0.08 | 0.05 | 0.12 | 0.06 | 0.01 | -0.02 | 0.12 | 0.04 | 0.07 | -0.06 | | 0.04 | p<0.0001 |

**Fig. 5 Inter-relationships between the PD-GIS subscales and mental health assessment.** Top left. Pearson's bivariate correlation r values. Only a few correlations were statistically non-significant at p < 0.05 two tailed and FWE corrected for multiple comparisons (underlined). Significant correlations were evident between measures of self-perceived COVID-19 pandemic impact, as captured in the PD-GIS subscales, and the mood self-assessment items. Top right. Canonical correlation analysis confirmed this relationship in a multivariate manner, with 7 statistically significant correlation modes as evaluated using Bartlett's approximate chi-squared statistic for $H(k)0$ with Lawley's modification, the largest with canonical r = 0.56 and the last mode significant at $p = 6.5e-26$. These results show that the effects of the pandemic on daily life have a significant component in explaining individual differences in standard mental health assessment measures. Source data are provided as a Source data file.

by one or more latent variables by applying principal component analysis (PCA) to PD-GIS responses (Supplementary Fig. 10). Permutation testing[18] indicated an optimal seven-component model. After varimax rotation, the first component had heavy loadings for questions pertaining to positive perception of more free time, less stress and reduced tiredness (Fig. 4 right). The second component represented questions pertaining to disruption of normal life and increased loneliness. Based on the item loading matrix, subsequent components were labelled 3: increased health concern; 4: positive outlook; 5: more conflict at home; 6: improved environment; 7: more time for people at home.

**Predicting standard mental health measures from self-perceived impact components.** Component scores from the seven self-perceived impact components were cross-correlated with the standard mental health assessment measures. The robustness of the relationship was corroborated by multivariate analysis with canonical correlation analysis (CCA), which produced seven statistically significant correlation modes (all p < 0.001), with small–large-scaled canonical r values (Fig. 5). Train–test analysis demonstrated that these substantial canonical correlations were not a consequence of overfit (Supplementary Fig. 11).

Bivariate correlations showed that the strongest relationships with anxiety for the PD-GIS components were health concerns followed by disruption of normal life. Depression and insomnia also correlated most strongly with disruption of normal life and health concerns. Irritability correlated with conflict at home. Positive PD-GIS components generally had small correlation effect sizes in the expected direction, e.g. participants indicating they had slept more and were less stressed and tired showed small benefits in terms of mental health scores.

**Population factors predict self-perceived pandemic impact.** Together the above results demonstrated the utility of the PD-GIS in providing a multi-dimensional assay of self-perceived

pandemic impact and the relevance to mental health measures. This enabled us to examine how population variables relate to pandemic impact. We used GLMs to examine the relationships of PD-GIS scores with sociodemographic and economic variables, home context, cohabitees and work arrangements (Supplementary Tables 17–24 for full results). In accordance with the mental health assessment analysis, age was among the most prominent predictors, having strong correlations with all seven self-perceived pandemic impact components. Examined at a yearly grain (Fig. 6), these differences spanned the medium to very large effect size range but were often non-linear. Older adults scored lower for more time and less stress and higher for increased health concerns. Younger people scored higher for disrupted daily life. Teenagers and early 20s scored higher for increased conflict at home. People of working age had less positive longer-term outlooks. People of older working age scored higher for perceiving improved environment and those of middle working age scored higher for having more time for people at home, i.e. despite having a similar likelihood of living alone to young adults.

In addition to age, the most prominent predictors of self-perceived pandemic impact were occupational status and cohabitees (Fig. 7). Healthcare workers benefitted the least from more time and less stress and reported greater increases in health concerns. This was balanced by reporting more positive outlook for the future and the least disrupted lifestyles. Furloughed people benefitted the most from more time and less stress, followed by home workers, but along with students, reported the most disrupted lifestyles. Disabled or shielded people had some of the most negative perceptions, reporting little benefit of more time and less stress, having the greatest increase in health concerns, and being least likely to report improved environment. Retired people also reported heightened health concerns and were least likely to report having more time for people at home, i.e. even having accounted for levels of living alone. Disabled or shielded and home workers scored highest in terms of more time for people. People living with small children also perceived

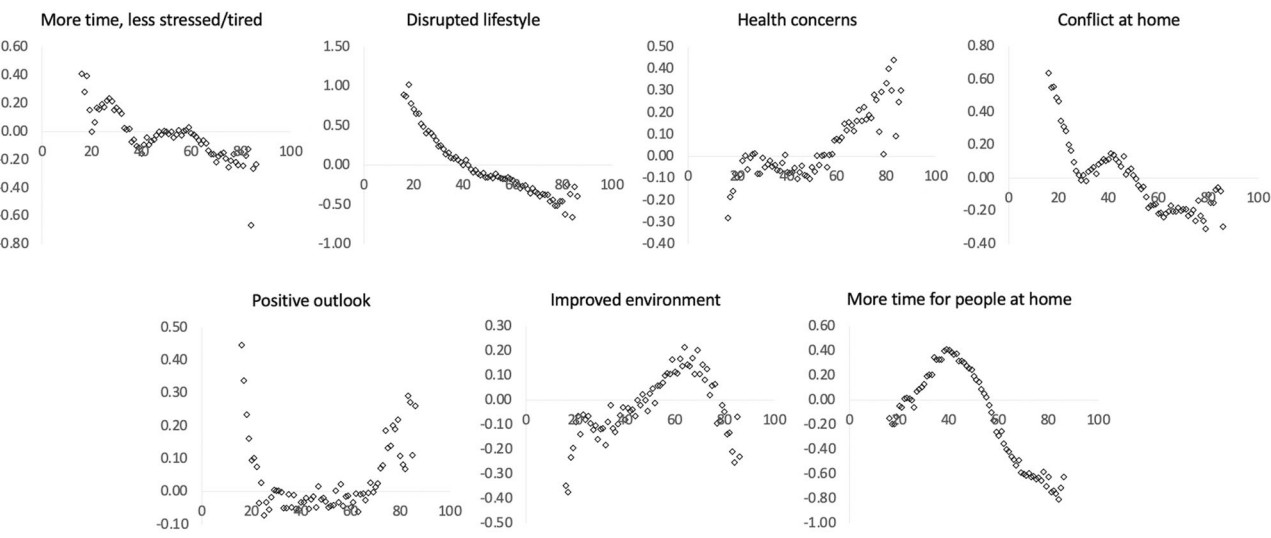

**Fig. 6 Self-perceived pandemic impact by age.** There were large associations between peoples' ages and self-perceptions of the ways that they had been affected by the pandemic (X axes = age, Y axes = component scores in SD units). Source data are provided as a Source data file.

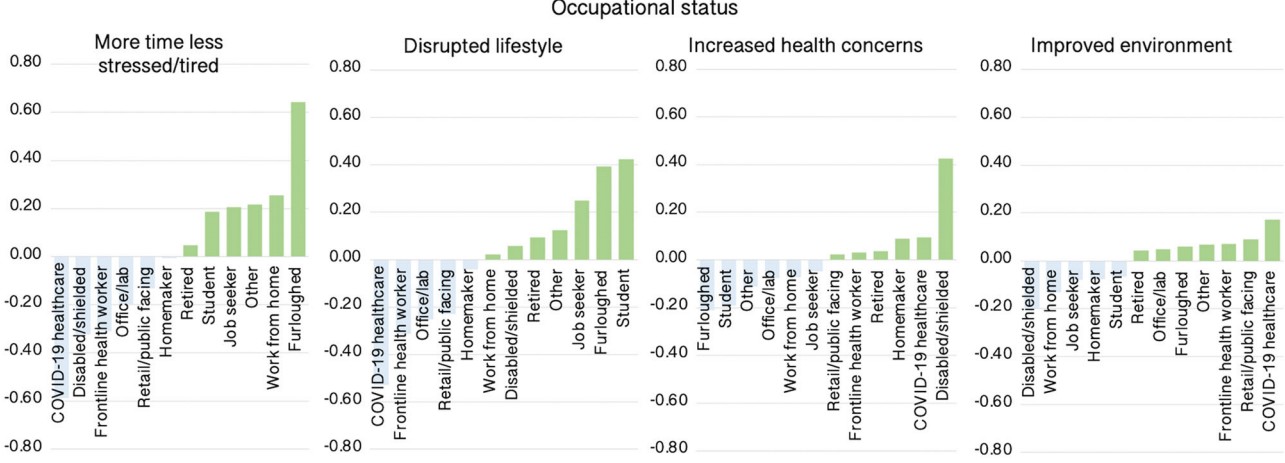

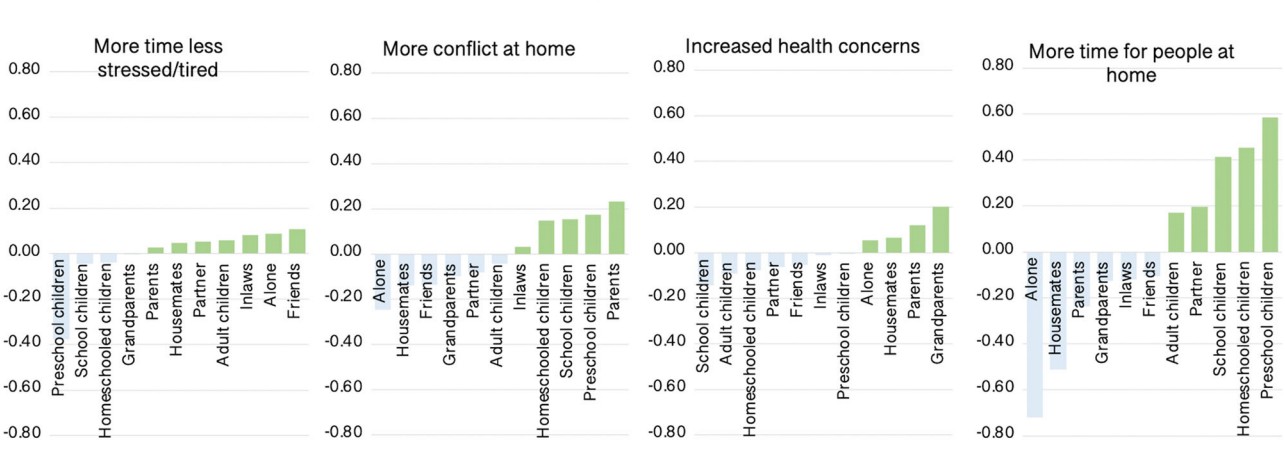

**Fig. 7 Self-perceived pandemic impact by occupational status and cohabitees.** There were medium–large-scaled associations between occupational status and cohabitees with self-perceptions of the ways that they had been affected by the pandemic. Y axes = mean centred GLM parameter estimates when predicting component scores (SD units). Source data are provided as a Source data file.

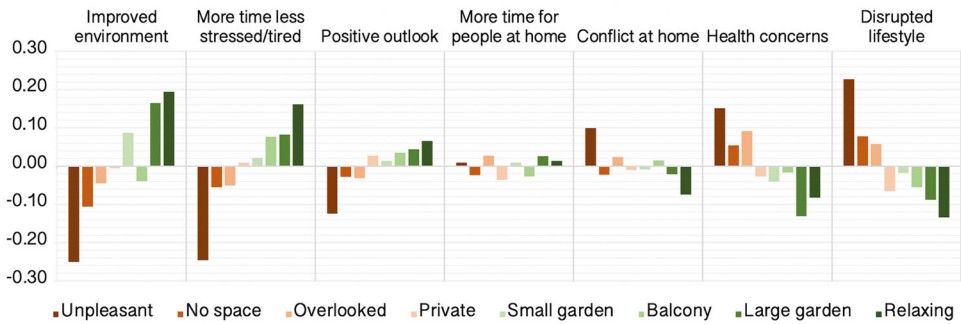

**Fig. 8 Self-perceived pandemic impact by outside space at home.** There were generally beneficial medium-scaled associations between quality of outside space people had at home and self-perception of the ways that they had been affected by the pandemic. Y axes = mean centred GLM parameter estimates when predicting component scores (SD units). Source data are provided as a Source data file.

disproportionate impact, scoring lowest in terms of more time and less stress and highest for increased conflict at home, with this somewhat balanced by more time for people at home. People living alone scored highest for disruption of lifestyle and, as expected, lowest for more time for people and conflict at home.

Small–medium relationships were evident for other population variables. Interestingly, type of house had small–negligible relationships with self-perceived impact scores, whereas quality of outside space predicted more positive report across many components, e.g. indicating more time and less stress, less disruption of lifestyle, less health concerns, less conflict at home and greater perception of improved environment (Fig. 8).

Higher education level predicted lower positive outlook. Males were less likely to report increased health concerns than females or people selecting other, females were more likely to report improved environment and people indicating 'other' had more positive outlook. Differences between ethnicities were generally in the small–negligible range, with the exception of health concerns, with people indicating Asian, Hispanic and Other scoring highest.

**Self-perceived pandemic impact in people with pre-existing conditions.** Next, we tested the prediction that people with pre-existing conditions would be more adversely affected. GLMs estimated differences in PD-GIS scores for people reporting established diagnoses of mental health and neurological conditions and conditions predisposing to COVID-19 vulnerability relative to controls (Supplementary Tables 25–31 and Fig. 9) after factoring out potentially confounding effects of the above population variables. There was high comorbidity between obsessive compulsive disorder (OCD) and anxiety and between anxiety and depression within the cohort. These participants were modelled as separate categories. Disorders with <100 representatives were moved to 'other neurological' and 'other psychiatric' categories. Small and statistically significant relationships were evident for disrupted lifestyle, which scored higher for people reporting diagnoses of depression. Notably, increased health concerns scored higher for people with pre-existing conditions that pre-dispose to COVID-19 risk, including diabetes, lung, heart and weakened immune system, but were greater still for people with anxiety disorders and OCD (~0.49 SD). People with attention deficit hyperactivity disorder (ADHD) scored marginally higher for conflict at home (~0.18 SD). There was a trend towards lower scores for improved environment across conditions, with this being greatest for people with bipolar disorder (~−0.22 SD) and people with Parkinson's Disease (~−0.25 SD).

**Role of technology use and personality traits in self-perceived pandemic impact.** Finally, linear modelling was applied using a robust train–test pipeline to quantify multivariate associations of

online technology use, personality traits and compulsivity with self-perceived COVID-19 impact after factoring out socio-demographic variables (full results in Fig. 10). Correlating predicted by true PD-GIS scores for the test data sets showed that a substantial proportion of the variance was explained by these traits, with the relationship being strongest for health concerns (mean $r = 0.32$, $p < 0.0001$) followed by disrupted lifestyle (mean $r = 0.24$, $p < 0.0001$) and then appreciation of improved environment (mean $r = 0.22$, $p < 0.0001$). Bivariate correlations showed that a combination of small-scaled correlations formed the basis of this multivariate relationship. The most prominent predictors related to negative PD-GIS components of disrupted lifestyle and more health concerns—specifically, personality traits including 'self-security' and 'conscientiousness'—were associated with reduced scores, whereas technology addiction (i.e. problematic usage of the internet), stress from technology and compulsivity (reward drive and rigidity) were associated with increased scores.

## Discussion
By analysing UK data at large scale collected from individuals immediately prior to the COVID-19 pandemic, and around the time of first UK lockdown, our study provides insights into the idiosyncratic ways that mental health and wellbeing have changed, including the relationship to self-perceived impact on everyday life and the population variables that modulate vulnerability and resilience.

On average, we observed only subtle differences in conventional mental health measures pre- to peak-UK lockdown. Anxiety rates showed the most pronounced increase. Depression rates were if anything slightly lower, while in parallel, people reported sleeping more and being less tired. Notably, prior studies in this area have provided mixed results, with some reporting subtle and others more pronounced pandemic impact[12,19,20]. Part of this incongruency may relate to differences in baseline calculation approaches, with a reliance on old historical data or baseline data collected after the pandemic was entrenched in many prior studies. The incongruency in prior work also likely pertains to the application of statistical models that do not account for the breadth of relevant population variables.

Age was the most prominent population variable accounting for pandemic impact. Older people showed disproportionate negative differences in standard mental health measures: e.g. greater increases in anxiety, less increase in sleep and concomitant reduction in tiredness, and in the case of depression, increases as opposed to decreases. Conversely, younger people on average already had higher anxiety and depression scores prior to the pandemic, and their scores remained higher mid-lockdown. These findings reinforce the message advocated in a recent

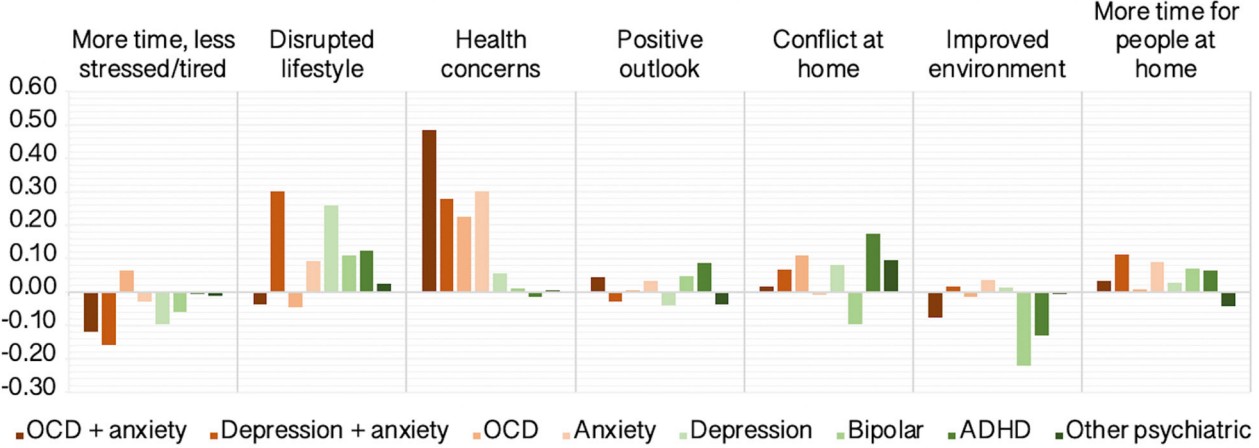

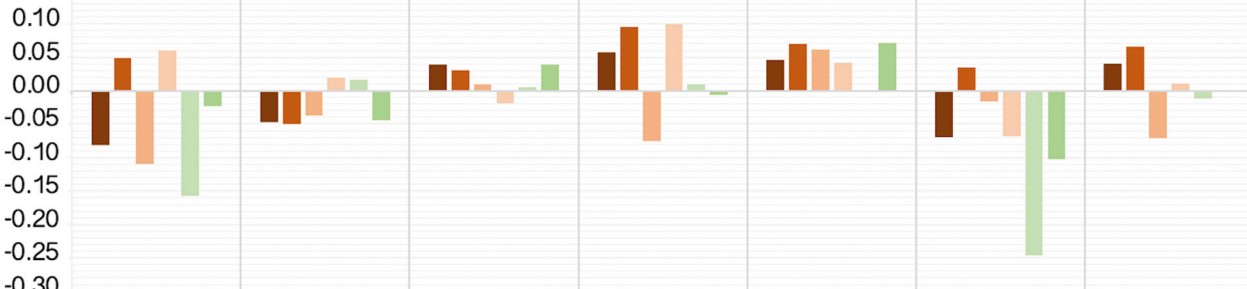

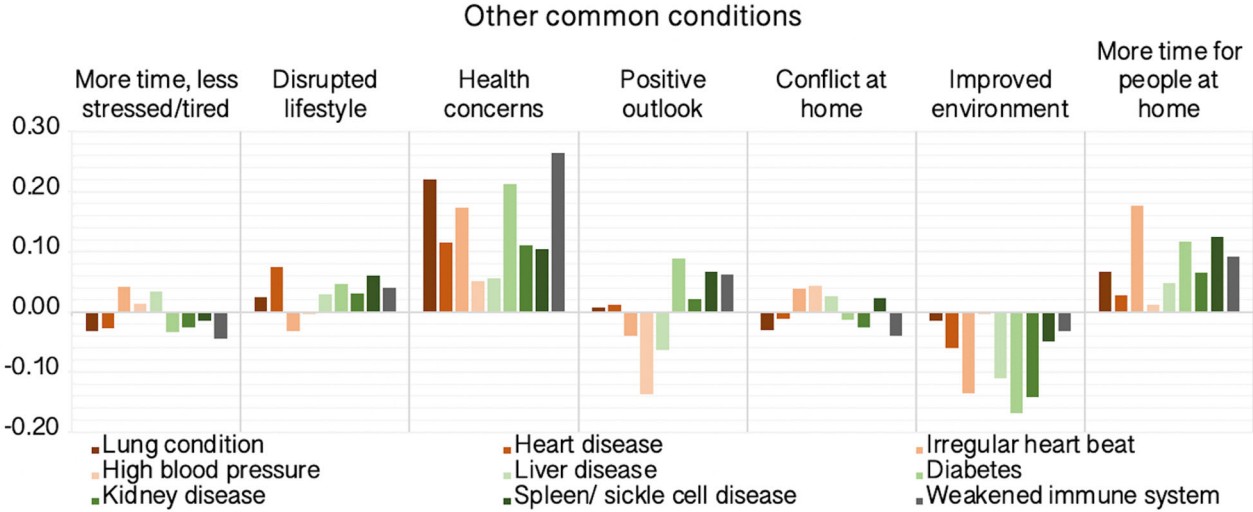

**Fig. 9 Self-perceived impact of the COVID-19 pandemic by pre-existing conditions.** Differences in PD-GIS component scores for people reporting psychiatric, neurologic and other pre-existing conditions relative to controls in SD units. Source data are provided as a Source data file.

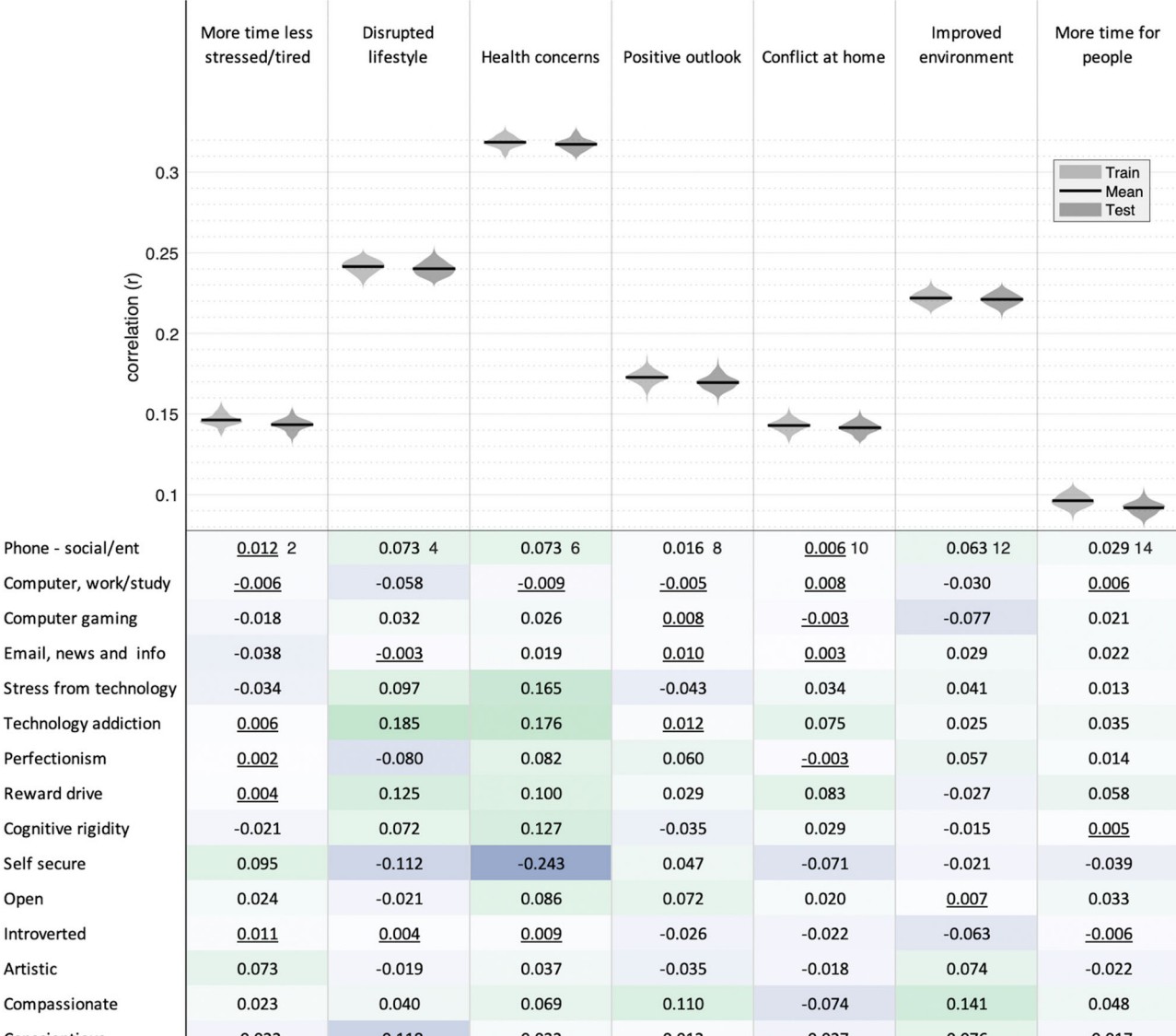

**Fig. 10 Correlation of trait and technology with PD-GIS component scores.** Top. Linear modelling showed that collectively emotion, compulsivity and technology-use traits predict substantial variance (all $p < 0.0001$ two tailed and FWE corrected for multiple comparisons) in dimensions of self-perceived pandemic impact, particularly for health concerns. Violin plots show the distributions of Pearson's correlations for 100 models trained on random 50% subsets of the data (light grey) and the correlations of predicted vs observed values when those trained models were applied to the remaining 50% of the data, to which it was naive, with means highlighted in black. Note the lack of overfit. Bottom. Bivariate correlations were mostly statistically significant and generally in the small range. Non-significant correlations at the two-tailed criterion of $p > 0.05$ after FWE correction for multiple comparisons are underlined. The strongest relationships were evident for (i) disrupted lifestyle, which correlated positively with technology addiction and reward drive and negatively with self-security and conscientiousness; and (ii) health concerns, which correlated positively with stress from technology, technology addiction, reward drive and cognitive rigidity and negatively with self-security. Source data are provided as a Source data file.

position statement[2] and elsewhere[21,22] that there is cause for concern about the mental health of both younger and older people during the pandemic. While considerable attention has been drawn to the impact of the pandemic in younger people, we believe that the disproportionate impact on older people has been overlooked.

Regarding interventions for mitigating the impact of the pandemic on mental health, useful insights can be drawn from how people perceive their everyday lives to have been affected. In this respect, results from the analysis of the PD-GIS scale are informative. First, the ways in which people consider themselves to

have been affected are complex and highly idiosyncratic. Seven principal components were required to capture variance in self-perceived impact. Second, the positive and negative dimensions not only segregated from each other but also explained substantial variance in the standard mental health measures, providing different potential targets for mental health interventions both during and beyond the pandemic. Finally, along with the mental health measures, the dimensions of self-perceived impact covaried with key population variables like age. Therefore, although some potential approaches to mitigating the mental health impact of the pandemic may be generic, it is likely that

others should be tailored to the individual. We believe that intersecting scales such as the PD-GIS with relevant demographic and other contextual variables provides a promising avenue for enabling such tailoring.

Notable among the broader positive associations was access to pleasant outdoor space, which was a marked statistical determinant of more positive self-perceived pandemic impact, relating to being less stressed and tired, fewer health concerns, a more positive outlook and a less disrupted lifestyle. These data are consistent with prior work in non-COVID-19 literature suggesting that access to green space is not only associated with mental health benefits and reduced stress[23,24] but also extend that association to resilience against the negative conditions brought about by the pandemic.

In terms of self-perceived negative impact of the pandemic, of particular note were the components 'increased health concerns' and 'disrupted lifestyle', as these had the strongest relationships with standard mental health measures. Age again featured prominently. It was clear that self-perceived negative impact came in different forms, e.g. for older adults—who had the highest health concerns—young adults and teens—who reported being the most disrupted and having greater conflict at home—and those of middle working age—who reported more time for people at home and improved environment but reduced positive outlook. These results highlight that age is an important factor to consider when attempting to design interventions to mitigate the pandemic impact on mental health.

Demographic characteristics, work, environment and social circumstances also all had robust associations with self-perceived pandemic impact, varying in scale from small to very large. Most notably, health workers showed very large differences to the broader population, being less relaxed and having less free time but also reporting better sleep and greater work engagement. These results are not only in keeping with disproportionate psychological effects of outbreaks in healthcare workers reported in previous literature on other infections[25], as well as initial data available regarding COVID-19[26,27], but also indicate that perceived benefits of such work are experienced by healthcare workers. Maximisation of workforce wellbeing among healthcare professionals is especially crucial not only during the pandemic[28,29] but also beyond it given the escalating backlog of more routine care being deferred due to the pandemic. People who had been furloughed were less stressed and tired compared with healthcare workers, whereas those with loss of income reported more life disruption compared with those who had retained their jobs. Finally, in terms of living circumstances, cohabiting with parents was associated with the largest increases in conflict at home during the lockdown, as was living with younger children.

In addition to revealing important differences in self-perceived pandemic impact contingent on demographic and vocational and home contexts, the study sheds light on mental health resilience and vulnerability in the general population. We hypothesised that the self-perceived impact of the COVID-19 pandemic would be associated with psychiatric and neurologic disorders[1,2] and dimensional traits[8]. Although, overall, people with psychiatric and neurologic conditions were less likely to report increased connectedness during the pandemic, these generalised associations were of small scale. More selective associations were observed for particular disorders: elevated health worries in anxiety disorders and OCD and increased conflict at home in adults with ADHD[30,31]. The pandemic outbreak for people with OCD may reinforce underlying health anxieties and serve to reinforce underlying belief systems; an initial study in OCD patients indeed reported worsening of symptom severity longitudinally following the outbreak of the pandemic[32]. Guidance to

help management of OCD during the pandemic is available[33]. A previous survey of parents reported worsening of their children's ADHD symptoms during the pandemic[34], which has been proposed to reflect the emotional dysregulation inherent to the disorder[30,35–37]. The observation here of self-reported increases in conflict in the home for people with ADHD accord closely with this view.

Personality, technology use and compulsivity traits were strongly interrelated and collectively accounted for substantial variation in pandemic impact. Most notably, people who reported having increased health concerns tended to have higher insecurity, cognitive rigidity, reward drive, technology addiction and stress from technology. A similar pattern of association was evident for people reporting more disrupted lifestyle. These results accords with the notion that certain traits are prominent in shaping resilience, whereas others engender vulnerability[38]. A crucial aspect of these data is that, whereas certain conventional personality traits have been previously related to resilience, including during COVID-19 lockdown[39], the roles of compulsivity and technology use have received very limited attention in this context. In part, this may be due to a historic lack of available, validated trans-diagnostic compulsivity instruments[40]. Here compulsivity (reward drive and cognitive rigidity)[41,42] were shown to associate with vulnerability, robustly associating with negative self-perceived pandemic impact.

Technology use bore important statistical relationships with positive and negative dimensions of self-perceived pandemic impact. Prior studies proposed that a subset of people develop problematic usage of online technology[43–46], and a consensus statement suggested that this would be especially impactful during lockdown[3]. Here we found that negative COVID-19 impact was strongly linked not with time spent using online technology per se but rather with maladaptive online behaviours. Relatedly, the most prominent technology stressor was reading news articles. Conversely, the benefits of using technology to stay connected were prominent in the questionnaire. Looking forward, one might infer that encouraging people to simply limit screen time could be counterproductive; more nuanced approaches that foster healthy online behaviours[47,48], while minimising maladaptive use[3,45,46,49], have relevance through and beyond the recovery phase of the pandemic.

The primary limitations of this study pertain to cross-sectional analysis, albeit within the context of a longitudinal study. We believe this limitation is mitigated by the large scale of data, the close matching of participant demographic profiles pre vs mid pandemic and the rigorous application of multivariate statistical analyses to take into account diverse potentially confounding population factors. They also are counterbalanced by the ability to compare data at scale just prior to the pandemic and midway through the first lockdown. A further limitation is the focus on self-report measures, although we note that this limitation applies to much pandemic-related mental health research, since face-to-face data collection exposes research participants and study teams to potential infection risks.

Finally, a critical consideration for any survey-based study is how representative the sample is of the general population. All sampling approaches, probabilistic or otherwise, have some degree of bias as participants must be accessible and motivated to respond. We believe a key strength of our study is its use of a large-scale non-probabilistic approach to capture baseline in the weeks immediately prior to the pandemic in the UK rather than relying on historical data or baseline data collected after the pandemic was already entrenched. We sought to maximise inclusivity through the high visibility to the general public of the BBC. Furthermore, the bias analysis showed only negligible scaled differences in mental health measures between those who did vs

did not choose to complete the PD-GIS scale. The resultant sample compares favourably to bioresource-based and random sampling studies in terms of inclusivity of minority groups and age range. More importantly, stratified, formally representative population samples are often constrained to one or two measures of interest, including those deployed to date in the COVID-19 context. In contrast, our inferences are based on analysis models where broad psycho-socio-demographic variables are accounted for, this being enabled by unusually large sample size. Future research should quantitatively compare in greater detail the relative biases present in mainstream media-based vs other sampling approaches.

Looking forward, we are recontacting this cohort at 6 and 12 months to plot change in the idiosyncratic impact of the pandemic and its aftermath[12,50], including more detailed assessment of identified vulnerable subgroups. We also note that the current paper focussed on data from people aged ≥16 years—we intend to examine data from younger people in separate work. It is vital that research examines the impact of the pandemic in younger people using this and related methodologies adapted for that purpose[2]. Regarding domains of self-perceived impact, the PD-GIS may not capture all domains that people find important in terms of pandemic impact. A future article will report topic modelling of free text reports of pandemic impact collected alongside the current data set.

In summary, our results demonstrate the value of measuring multiple dimensions when quantifying pandemic impact on mental health and highlight the necessity of incorporating the broad psycho-socio-economic context when seeking to understand, predict or mitigate such impact. The largest associations explaining mental health during the pandemic and self-perceived pandemic impact on daily life related to age, occupation and home context and external space, followed by medium associations with personality traits, compulsivity and maladaptive technology use, and smaller but still notable associations for mental health and neurological disorders and demographic characteristics. The ways in which different vulnerable groups report being impacted by the pandemic can provide rich information to help guide future resource provision, research and clinical interventions, in order to maximise outcomes and minimise untoward effects of the COVID-19 pandemic as it continues to manifest, especially in periods of lockdown or social restrictions. To this end, we are making our data set available for detailed analysis by the broader research community.

## Methods

**Recruitment.** Starting from December 26, 2019, participants were recruited to the study website, where they completed cognitive tests and a detailed questionnaire. The sampling approach was large scale and non-probabilistic. Specifically, to maximise visibility and inclusiveness articles describing the study were placed on the BBC2 Horizon, BBC homepage, BBC News homepage and circulated on mobile news meta-apps from January 1, 2020[51] in the UK. To maximise representativeness of the sample, there were no inclusion or exclusion criteria. Analyses here exclude data from participants aged <16 years, as they completed a briefer questionnaire, and those who responded to the questionnaire unfeasibly fast (<4 min). Cognitive test data will be reported separately. The study was approved by the Imperial College Research Ethics Committee (17IC4009) and participants provided informed consent prior to starting the test. Data were collected from individuals at a particular time point, rather than data being collected serially from the same individuals.

**Data collection.** Data[52] were collected via our custom server system, which produces study-specific cognitive testing and survey websites (https://gbws.cognitron.co.uk) on the Amazon EC2. Questionnaires and tests were programmed in JavaScript and HTML5. They were deliverable via personal computers, tablets and smartphones. The questionnaire included scales quantifying sociodemographic, lifestyle, online technology use, personality and mental health (Supplementary Methods)[42,53–56]. Participants could enrol for longitudinal follow-up, scheduled for 3, 6 and 12 months. People returning to the site outside of these time points were

navigated to a different URL. On May 2, 2020, the questionnaire was augmented—in light of the COVID-19 pandemic—with an extended mood scale and an instrument comprising 47 items quantifying self-perceived effects on mood, behaviour and outlook (PD-GIS). PD-GIS specifically asked about changes due to the pandemic (increases and decreases). Questions regarding pre-existing psychiatric and neurological conditions, lockdown context, having the virus and free text fields were added. This coincided with further promotion via BBC2 Horizon and BBC Homepage.

**Data processing and statistical analysis.** Analyses were conducted in MATLAB R2020a. Participants with missing data were retained as some questions were contingent on others; therefore, observations per analysis vary with data availability. The questionnaire was organised into the following scales: demographics and lifestyle, online technology use, mood, personality, compulsive traits, and pandemic impact. Where appropriate, scales were summarised in the following steps. Agree–disagree and frequency items were filtered for missing data casewise within scale, converted to numeric, rank transformed to normality and subscale scores estimated using PCA. Components with eigenvalues >1 were varimax rotated and component scores estimated by regression. While there are other methods for determining factor fit, this is widely used in the literature and is convenient. PCAs are reported in Supplementary Figs. 12–16 and Supplementary Tables 32–41.

Cohort demographics were segmented into pre-pandemic, early-pandemic, and mid-pandemic epochs (Fig. 1). GLM tested how population variables statistically predicted differences in mood, anxiety and sleep between the pre-pandemic and mid-pandemic epochs. To enable this, responses were converted to a numeric scale, ranging as follows: 0 (never), 1 (rarely), 2 (once or twice a week), 3 (several times a week), 4 (every day), 5 (hourly), and 6 (more often). These two epochs were analysed because they had large sample sizes and comprised participants with similar demographic distributions, whereas the early-pandemic epoch comprised generally younger aged individuals.

Subscales of the pandemic impact instrument (PD-GIS; Supplementary Fig. 10) quantified self-perceived impact across seven psycho-socio-economic dimensions. These were identified by applying a permutation-based approach, whereby eigenvalues for the true data were cross-compared to 10,000 permutations in which the linkage across measures was destroyed[18]. Components with eigenvalues >95% of the permutations at the corresponding index were retained. The rotated component scores were cross-related to the mental health self-assessment in a multivariate manner using CCA, which finds linear combinations of the two multivariate matrices that have maximum correlation with each other[57], and in a bivariate manner using Pearson's correlation.

GLM determined the relationship of sociodemographic variables, home context, cohabitees and work arrangements to the seven components of the PD-GIS. Further GLMs examined carers and people reporting psychiatric and neurologic diagnoses (for groups with $N > 90$ members) with the sociodemographics factored out.

Due to the expected high shared variance between online technology use, personality traits and compulsivity, their multivariate relationships with the PD-GIS were quantified using a linear regression support vector machine regression model[58] in MATLAB, specifically, the fitrlinear function, which by default uses support vector machines with a ridge penalty and optimises using dual stochastic gradient descent. To evaluate generalisability of the model and degree of overfit, a train–test pipeline was applied, whereby the model was trained by fitting the trait measures onto the scores for each individual PD-GIS component with 50% of the data, and the $r$ value then estimated between predicted and true data when the trained model was applied to the same data, and the other 50% of the data, to which it was naive. This process was repeated 100 times with different random splits and the mean and SD correlation value was reported.

**Reporting summary.** Further information on research design is available in the Nature Research Reporting Summary linked to this article.

## Data availability

All questionnaires and data, both raw and filtered for analysis, are available via the UK Data Service: COVID-19 impact data set: Great British Intelligence Test, 2020 https://reshare.ukdataservice.ac.uk/854451/. No limitations are applied on use of the shared data. A key detailing the location of data and models for main display items is provided in the Supplementary Information. Source data are provided with this paper.

## Code availability

Native MATLAB functions were used to generate all statistical models in this study and these are available via the UK Data Service: COVID-19 impact data set: Great British Intelligence Test, 2020 https://reshare.ukdataservice.ac.uk/854451/.

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

## Acknowledgements

This study was conducted in collaboration with BBC2 Horizon. The study was supported by the UK Dementia Research Institute and Biomedical Research Centre at Imperial College London. Technology development was supported by EU-CIG EC Marie-Curie CIG and NIHR grant II-LB-0715-20006 to A.H. E.S.'s role was supported by MRC grant MR/R005370/1 to A.H. W.T. is supported by the EPSRC Center for Doctoral Training in Neurotechnology under supervision of A.H. This research was funded in part by Wellcome [110049/Z/15/Z and 110049/Z/15/A] (Grant to S.R.C.). For the purpose of open access, the author has applied a CC BY public copyright licence to any Author Accepted Manuscript version arising from this submission. M.A.M. is in part supported by the National Institute for Health Research (NIHR) Biomedical Research Centre at South London and Maudsley NHS Foundation Trust and King's College London. The views expressed are those of the author(s) and not necessarily those of the NHS, the NIHR or the Department of Health and Social Care. We would like to acknowledge COST Action CA16207 'European Network for Problematic Usage of the Internet', supported by COST (European Cooperation in Science and Technology), and the support of the National UK Research Network for Behavioural Addictions (NUK-BA).

## Author contributions

A.H. designed and instigated this study, arranged promotion of the study URL, oversaw design of the questionnaire and programming of assessment software, conducted all statistical analyses, created figures and authored the article. P.J.H. contributed to the design of the questionnaire and the study, programmed the server-side software, oversaw data collection and curation and edited the article. E.S. contributed to the design of the study questionnaire, developed custom visualisation software for the project and edited the article. M.A.M., K.I. and J.E.G. contributed to the design of the study questionnaire and edited the article. W.T. contributed to the design of the questionnaire and the running of the study, programmed the custom questionnaire software with A.H., contributed to data collection and curation and edited the article. S.R.C. contributed to the design of the study and of the study questionnaire, provided input on the analyses and co-wrote all versions of the article.

## Competing interests

S.R.C. previously consulted for Promentis. He receives honoraria for journal editorial work from Elsevier. J.E.G. has received research grants from the T.L.C. Foundation for Body-Focused Repetitive Behaviors, Biohaven, Promentis and Avanir Pharmaceuticals. M.A.M. has received grant income from Takeda Pharmaceuticals, Johnson & Johnson and Lundbeck. A.H. is owner and founder of Future Cognition Ltd. and H2 Cognitive Designs Ltd., which develop custom cognitive assessment software for other university-based research groups. P.J.H. is the owner and co-founder of H2 Cognitive Designs Ltd. The authors report no other conflicts of interest.
