## [Peer Review File · Nature Communications]

Reviewers' comments:

Reviewer #1 (Remarks to the Author):

This is a review for manuscript NCOMMS-20-28403-T. The paper uses data-driven methods to describe population-level and individual differences in changes in mental health, the varied impacts of the pandemic, and participant-generated coping strategies.

There is a lot to like about this paper. The extremely large and sociodemographically diverse sample allows for precise estimation of effect sizes and meaningful investigation of differences between subgroups (including minority groups). The study includes a large and comprehensive set of predictors (e.g., age, gender, vocations, circumstances, personality traits). The large set of outcomes also facilitated nuanced insights about the mixed implications for different indicators of mental health (e.g., on average, anxiety increased but tiredness decreased and people got more sleep between pre- and mid-pandemic). Finally, I greatly appreciated the paper's focus on understanding the differential impacts of the pandemic on different groups (rather than focusing on average trends). All in all, this paper contributes timely insights that could be used to inform mental health interventions during these times, and also has the potential to contribute to knowledge on mental health resilience, coping, and personalized interventions outside of a pandemic context.

As detailed below, my main concerns are about the adequacy and effectiveness of the reporting.

Major concerns

1. There is a *lot* of data in this paper. In its current form, the density of this paper and its supplement are overwhelming, and it is unclear what insights/key messages readers should take away from it. The majority of the results are buried in a series of difficult-to-read tables in the supplement. The impact of the paper would remain limited if the results remain this difficult to navigate and make sense of. At the very least, the massive tables in the supplement should be reported as separate tabs in an Excel spreadsheet. This would make it easier to filter and sort the results. An even bigger contribution would be if the authors created an interactive web app that would allow people to see relevant results. For example, in the section on recommended advice, the manuscript only reports a few select differences in the advice given by people in various subgroups, and only reports the advice prevalence for all 50 topics for retired people relative to all others (Figure 7b). It would be extremely useful if researchers and the general public alike were able to go to a website, select the group of interest (or perhaps even an intersection of groups), and see the advice topic prevalence for that group (e.g., in a similar format as Figure 7b). This would also be applicable for the topics on the positive and negative impacts of lockdown, to help researchers and policymakers better understand the differential impacts for different groups.

2. I evaluated the methods and statistical analyses to the best of my ability, but this was sometimes difficult because 1) not enough detail was provided, or 2) some of the statistical methods might need

some explanation to be accessible to a general social science audience (I am writing from the perspective of a psychologist with substantial training in advanced quantitative methods but limited knowledge of LDA and machine learning techniques). Here are some points that I thought needed clarification:

- a) It is unclear whether any of the same participants were followed over time between the pre- and mid-pandemic waves, or whether these samples were completely independent. Were the 74,830 participants who responded to the PD-GIS-11 a subset of the 110,118 mid-pandemic participants? Were the 8,680 early-pandemic participants included in any of the analyses?
- b) p. 5: “Differences in mood assessment scores were calculated for the densely sampled and closely matched Mid vs Pre-Pandemic epochs after factoring out age, sex, handedness, education level, first language, country of residence, occupational status and income”. What is meant by “densely sampled” vs. “closely matched”? Does this mean that there was another version of the analyses that involved comparing demographically-matched samples? Or does it just mean that after factoring out the various controls, the two samples became closely matched? In any case, it would be helpful to see a supplemental table with statistical comparisons and effect sizes for the differences between the two samples. It is difficult to gauge possible differences from Figure 1e.
- c) p. 5: I am confused by the rationale for calculating mood scores separately for each of the 31 days pre- and mid-pandemic, then running 31 comparisons, then reporting an average difference based on 31 comparisons. This seems to add unnecessary complexity—why not just compute an average mood score for the pre-pandemic epoch and post-pandemic epoch, and compare those?
- d) Figure 2: What are those colorful squares in the right panel? This needs more explanation. I also don’t understand why the X scale of the left panel is in SD units. What were the scores standardized with respect to? I strongly believe that it would more interpretable to know the level of agreement with each statement on the raw metric (i.e., x axis should be from 1 = strongly disagree to 5 = strongly agree).
- e) p. 8: I am not familiar with canonical correlation analysis, and don’t think the average social science reader would be either. Can you add a brief explanation of what this tells us? In Figure 3b, why are the last two rows empty? Does that mean that there was no correlation identified by the canonical correlation analysis?
- f) I did not understand what was going on in Figure 6, or the canonical correlation analysis that led to this. What are the two modes?
- g) Figure 7a would be more informative if it contained some information about the relative prevalence of the topics.
- h) For the advice topic analyses, why were t statistics and p values reported instead of standardized effect sizes (both in text and in Figure 7b and the supplement)? This was strange, given the previous emphasis on effect size rather than p values. It makes it difficult to know how large the relative differences were.
- i) Basic details about LDA analyses were missing (and further details were not in fact given in Supplement 4). At the very least, how were data tokenized (e.g., only words, or also 2- or 3-gram phrases)? What were the α and β parameters? How were multiword collocations identified (if applicable)? How were topic scores calculated? Did the final sets of 5 negative topics, 10 positive topics, and 50 advice topics capture ALL topics that were identified by LDA, or were there some uninterpretable

topics that were excluded?

j) Most of the supplemental tables and graphs need much more explanation (e.g., details of statistical analyses, x and y axis labels, whether coefficients were standardized) than what is currently provided.

3. Based only on the Kaiser rule of including components with eigenvalues > 1 (which has some problems, see <https://stats.stackexchange.com/questions/253535/the-advantages-and-disadvantages-of-using-kaiser-rule-to-select-the-number-of-pr>), the authors extracted 11 components from the pandemic impacts questionnaire. Given that so many of the subsequent analyses involve these resulting component scores, I would have expected to see a more thorough exploration of the dimensionality of this questionnaire, using different rules for determining the number of components to extract (e.g., visual inspection of the Scree plot, MAP test, parallel analysis, very simple structure criterion), as well as a consideration of the face-validity/interpretability of the resulting components.

To be fair, most of the components reported in Figure 2 seem to make sense, but there are some exceptions. For example, a) "Grieving" doesn't seem to fit with the "Concerned with health" dimension, b) might "More in touch with loved ones" belong in the "More time for people" or "More time, less stressed/tired" component?, c) is the isolated & disrupted component coherent (or would they perhaps be better summarized by two distinct components)?, and d) does "appreciating simple things in life more" really belong in "Improved environment"? In any case, it would be helpful to know how consistent the number and interpretability of the components are based on different criteria.

It would also be nice to see some discussion of how these 11 dimensions conceptually map (or don't) onto the 5 negative impact and 10 positive impact topics identified via LDA. Also, I could not find Supplements 4b and 4c (mentioned at the bottom of p. 13).

Relatedly, I request that the authors report the full text of the Pandemic General Impact Scale in the supplement. This is described in Supplement 1, but it would be helpful to see the exact wording that participants saw. For example, it is unclear what "improved natural environment" means.

Minor points

4. Data and analysis scripts were not available. For transparency, I encourage the authors to consider uploading the data and analysis scripts required to reproduce the results reported in this paper to an online repository such as the Open Science Framework. The reporting summary states that data are freely available, but where?

5. In the introduction, it is stated that "addressing these knowledge gaps during the pandemic provides a unique opportunity to understand the nature and psychological basis of mental health resilience and vulnerability in the general population" (p. 2). I was hoping to see greater discussion of how the current findings or the general methodological approach can contribute to knowledge about mental health beyond a pandemic context. For example, some more general lessons might be that 1) the effects of any given stressor/life event might have differential impacts depending on sociodemographic variables,

context, and personality traits, and 2) topic models of open-ended text responses could be used to identify promising personalized interventions more generally (not just for coping with a pandemic)—to me, this seems like one of the most exciting contributions of this paper (which is why I suggested making these data more easily-interpretable in point 1 above). The discussion would be improved with greater contextualization of the results within the broader literatures on personalized mental health interventions and the use of NLP for understanding mental health.

6. Here are some minor points of clarification/unclear sentences/typos I detected:

- a) p. 2: What is meant by a “many-to-many mapping relationship”?
- b) p. 3: “This resulted in 215,886 Pre-pandemic, 8,680 Early-pandemic and 110,118 Mid-pandemic datasets in the reported analyses”. It seems weird to refer to participants as “datasets”.
- c) Figure 1 caption: “Sociodemographically *mediated* differences” implies that the differences in population mood scores from pre- to mid-pandemic are *because* of differences in the sociodemographic composition of the two samples. “Sociodemographic correlates” would be more accurate.
- d) Figure 1c: needs a key or note in the figure caption to specify which colors correspond to pre- and post-pandemic.
- e) p. 5: “When analysing large-scale data, very small effects will tend to have highly significant statistical values. Therefore, the better measure of significance is effect size.” The difference uses of “significance” is confusing here. I would change the second “significance” to “practical importance”.
- f) p. 5: “Increases in depression was lower for younger adults and higher for older adults (-0.2SDs to 0.1SDs).” This sentence is unclear: did younger adults decrease in depression?
- g) Figure 5a-b: “Change in” implies change between pre- and mid-pandemic, but the impact scale was only administered at one time point. “Change in” should be deleted.
- h) Figure 3a: y axis needs labels or numbers—it’s difficult to cross-reference.
- i) Figure 3b: p cannot be .0000. Change these to $p < .0001$.
- j) p. 12: “mediated along” should be “correlated with”. “mediation” implies a process model with X explaining Y via mediator M.
- k) p. 13: “What has been most POSITIVE about the lockdown” was repeated twice. The second occurrence was supposed to be the question about coping.
- l) p. 16: Speaking as a personality psychologist, extraversion should definitely be spelled with an a (extraversion), not extroversion. This is the scientifically accepted spelling. See <https://blogs.scientificamerican.com/beautiful-minds/the-difference-between-extraversion-and-extroversion/> for a deep dive on the origins of the misspelling.
- m) Page 3 states that people who completed the questionnaire < 6 minutes were excluded, whereas page 18 defines “unfeasibly fast” as < 4 minutes. Which is correct?

Reviewer: Jessie Sun

Reviewer #2 (Remarks to the Author):

This manuscript reports a study based on a large-scale predominantly national survey. The topic is clearly timely and the study has been conducted according to advanced and methodologically sound approaches, including a sophisticated analytic plan. The reporting of the study is generally good and clear.

I do not have any major concerns, only a few suggestions:

Abstract: This is not fully clear to me; could this be rephrased?

Introduction:

There are currently a number of ongoing Covid-19 related surveys in the UK, aimed at assessing the mental/physical impact of the pandemic. The author may want to cite them (or at list the main ones) highlighting the strengths/weaknesses of their sample in relation to these existing ones.

The discussion should highlight the limitations of convenience samples (<https://pubmed.ncbi.nlm.nih.gov/32502467/>)

Reviewer #3 (Remarks to the Author):

This paper presents data-driven analyses from a unique and incredibly rich dataset that, notably, includes mental health measures pre- and post-pandemic! Given the unprecedented changes to life during Covid-19, I thought the decision to adopt a data-driven approach was warranted. However, I also felt that the authors had decided to report a bit too much to yield a coherent report. There is nothing wrong, of course, with reporting various analyses with different outcome measures, but this approach seemed to have resulted in insufficient detail of many of the specific analyses, so much so as to prevent readers from understanding what the analyses mean. It does not help that the authors go from analyses to analyses and from outcome to outcome with very little integration and explanation of how each piece fits together. The omission of important information varies from the cosmetic (e.g., failing to indicate what each color signifies in figures) to the fundamental (failing to specify the design: between or within-subjects). Thus, I am keeping my review fairly short, only highlighting some of the issues with the presentation and the analyses of this incredible dataset. That said, I do hope to see a revised version that is more detailed, organized, and ready for the limelight!

1. Many of the graphs are difficult to interpret due to insufficient information provided in the legends. For example, Figure 1c presents anxiety over 31 days pre and post-pandemic. OK, but which color indicates which period? Is the reader supposed to infer those visually from Figure 1a? But if that is so,

why are the same colors (blue and red) used to signify a completely different thing in Figure 1e (increase or decrease)? And again, the reader should not be left to infer that blue means a decrease, and red means an increase. The figures and their legends should be interpretable without having to read the entire paper. In Figure 4c, is White the comparison category? Also, why are d and e reported in the legend but not shown in the figure? Obviously, these are cosmetic issues, but they are also pervasive. As a reviewer, it is not my job to be a copy-editor, so I will not go on in detail about all the issues with the figures here, but the feedback is clear: Make your figures interpretable.

2. Though never explicitly specified, I am assuming that the authors are using a between-subjects design, meaning they are comparing different groups pre and post-pandemic. That said, some of the writing seems to suggest within-subjects design: “Six mental health measures were calculated and compared separately for people who were sampled at each of 31 days post-launch, effectively providing 31 independent replication analyses.” If the analyses are indeed between-subjects, then these are not really pre–post analyses, which typically implies a repeated, longitudinal design. We then must consider whether the participants in the two groups were randomly assigned to those groups, which is not the case. Finally, we must consider how those two samples might be different from each other (which might account for any differences that are observed). And the list of such factors is likely long: from the way participants were recruited to self-selection. For example, people who were more anxious to begin with might have been more likely to participate in a survey about COVID. Statistically controlling for a few factors cannot isolate COVID-19 as the direct causal factor. Yet, the analyses are interpreted exactly as a result of COVID.

3. I did find the moderation analyses by demographic factors to be most informative and interesting (Figure 1e). I wish the authors had explored these changes in anxiety, depression, and other outcomes measured with validated scales more in terms of other factors, such as current living situation, personality, and preexisting mental disorders, more—the same way they explored it with for the perceived impact outcome. I also found it confusing that the authors talk about demographically “mediated” effects. Mediation implies that X causes M, which in turn causes Y. This is not the case here as COVID-19 does not cause somebody to be man or woman, for example. These are moderation analyses, whereby the effect of COVID on Y depends on various demographic factors, M (the moderators).

4. I did find the analyses with the perceived impact to be interesting and informative. But while the authors acknowledge early on that these variables are capturing perceived impact, this is quickly lost in the deluge of analyses and by Figure 5, the authors proudly announce in the figure caption that they are showing “Impact” of Covid-19 as a function of mental disorder. Though this may seem like another cosmetic suggestion, it is a serious concern. The authors are effectively claiming—with little evidence—that they have studied the impact of COVID-19, when all they are showing is what people, in completely retrospective reflections, perceive as factors that have impacted them during COVID. The limitations of retrospective perceptions in making conclusions about actual impact are well documented and I will not elaborate further.

5. The discussion is another missed opportunity to integrate the findings and outline the limitations of the design and the analyses.

6. The authors casually state that “as a quality control people who completed the questionnaire too

rapidly (defined as under 6 minutes) were excluded.” This seems quite arbitrary. When was this decision made? Was this decision preregistered? Was it at least made before looking at any of the analyses? How did the authors decide on 6 minutes?

In sum, while data-driven analyses of such rich data have enormous potential to inform our understanding of who is most impacted by Covid-19, adopting a data-driven approach does not mean quickly reporting a multitude of analyses without sufficient detail and little integration or interpretation. And even the fanciest analyses and richest datasets can still not make up for careful, thoughtful interpretation of what the results mean and what they do not.

Reviewer #4 (Remarks to the Author):

This review is for "Dimensions and modulators of behavioural and mental-health change during the Covid-19 pandemic" (NCOMMS-20-28403-T). Overall, there are several things to like about this manuscript. It is generally well-written and timely, and both factors work to its advantage. Ultimately, my main critiques around this paper are really just centered in on the question of "what is the content, scale, and scope of this work's contribution to the literature?"

I found my main question somewhat difficult to answer. Much of my struggles here revolved around the fact that I didn't feel like there was really enough information to evaluate everything that was going on here. In many ways, and in many places, the manuscript feels a bit like a "rush to the finish line", so to speak. For many of the analyses, I found that the statistical reporting was rather incomplete, with a lot of the reporting feeling like there's an implicit statement of "please just trust that we were thoughtful and rigorous in our analyses". I don't doubt that the authors were, but given the size, source, and overall nature of the data, I would have expected perhaps more nuance and clarity about everything that's going on here. For things like the PCA and LDA, I didn't really see any evidence for the statistical decisions made, which is particularly necessary given their unsupervised nature (i.e., one can arbitrarily apply numerous criteria to determine the number of components, topics, etc.). Additionally, there simply wasn't enough information to really unpack these intermediate steps, let alone the results taken on the data.

Given the stated goals of the paper, I was somewhat surprised by the fact that, for the most part, the presentation of the data and results leaned towards the cursory and descriptive side of things, without really unpacking the meaning and nature of the lockdown specifically on the participants' psychology. Much of what was reported here was not particularly surprising (not that the findings need to be, or should be, in order to be reported/published). But, I felt that this work was fairly disconnected from just about every relevant literature there is. Out of 20 references, at least 50% were specific to COVID/coronavirus. However, the health psych literature broadly defined, large swaths of the social psych literature, the and massive amounts of psychopathology literature are relevant, and based on everything that we know from each of these rich fields of study, I don't think that there is anything being reported here that wouldn't be predicted and hasn't been shown, to some degree or another, in smaller samples (in many cases, still very large samples, just not as large as this one). Even the general psychometrics literature would probably chip away at much of measures here, citing problems of shared method variance, the extensive reliance of self-reports, and so on.

All of this said, I think that there remains plenty of value to this work, but I'm not sure if it really is coming to the surface or shining in the ways that it could. I would ask that the authors provide a vastly more detailed and extensive set of supplementary materials / data availability / code availability than what I found for this work (unless I'm missing something critical) so that reviewers could really dig in and do a deeper evaluation of the work that has been done here. I would also ask that the authors perhaps revisit their framing of the work being done here and think about what boundaries might be placed on the findings giving some of the not-insignificant psychometric considerations of the data as-is, and as it has been collected. And, as noted above, I would love to see this work much

more deeply connected with the relevant literatures, linking up to relevant findings that have been discussed elsewhere (particularly in health psych, but much beyond this as well), and discussing/demonstrating how this work builds on past research and brings new things to light that we might not already understand about factors such as individual differences, social media, relationships, demography, and health outcomes

General response to reviewers

We are grateful to the reviewers for their extensive, encouraging and helpful comments on our manuscript. We note that drafting this manuscript has been a remarkable challenge due to the unique nature of the dataset, which is extremely large and has unusual scope in terms of findings.

To clarify, our original strategy was to introduce this dataset, cover some of the more salient and timely findings from a data driven / holistic perspective, then make the dataset available for the broader research community to examine in greater detail than is possible in any one article. In retrospect, we covered too much ground with insufficient explanation.

This further submission has been heavily revised. We have worked hard to address all of the comments in detail, including complete reanalysis of the entire dataset (with addition of some new participants). Extensive restructuring has been undertaken, including more explanation of analyses and rationale/motivation, and more discussion of the implications of the results.

We believe that our results are highly relevant at this time as we move towards a second Covid-19 wave. Reflecting this urgency, our data already are available via the UK data service to the broader research community, who will no doubt wish to analyse certain aspects in greater detail. Analysis code will be made available at the point of publication. We hope that you consider our revised article suitable for publication and look forward to your response.

Yours sincerely,

Adam Hampshire (on behalf of all the authors)

Response to Reviewer #1

This is a review for manuscript NCOMMS-20-28403-T. The paper uses data-driven methods to describe population-level and individual differences in changes in mental health, the varied impacts of the pandemic, and participant-generated coping strategies.

There is a lot to like about this paper. The extremely large and sociodemographically diverse sample allows for precise estimation of effect sizes and meaningful investigation of differences between subgroups (including minority groups). The study includes a large and comprehensive set of predictors (e.g., age, gender, vocations, circumstances, personality traits). The large set of outcomes also facilitated nuanced insights about the mixed implications for different indicators of mental health (e.g., on average, anxiety increased but tiredness decreased and people got more sleep between pre- and mid-pandemic). Finally, I greatly appreciated the paper's focus on understanding the differential impacts of the pandemic on different groups (rather than focusing on average trends). All in all, this paper contributes timely insights that could be used to inform mental health interventions during these times, and also has the potential to contribute to knowledge on mental health resilience, coping, and personalized interventions outside of a pandemic context.

We thank the reviewer for their positive comments.

Major concerns

*1. There is a *lot* of data in this paper. In its current form, the density of this paper and its supplement are overwhelming, and it is unclear what insights/key messages readers should take away from it. The majority of the results are buried in a series of difficult-to-read tables in the supplement. The impact of the paper would remain limited if the results remain this difficult to navigate and make sense of. At the very least, the massive tables in the supplement should be reported as separate tabs in an Excel spreadsheet. This would make it easier to filter and sort the results. An even bigger contribution would be if the authors created an interactive web app that would allow people to see relevant results. For example, in the section on recommended advice, the manuscript only reports a few select differences in the advice given by people in various subgroups, and only reports the advice prevalence for all 50 topics for retired people relative to all others (Figure 7b). It would be extremely useful if researchers and the general public alike were able to go to a website, select the group of interest (or perhaps even an intersection of groups), and see the advice topic prevalence for that group (e.g., in a similar format as Figure 7b).*

This would also be applicable for the topics on the positive and negative impacts of lockdown, to help researchers and policymakers better understand the differential impacts for different groups.

We thank the reviewer for these helpful suggestions. Writing this article has been a singular challenge due to the unique scale of data and scope of findings that can be drawn from it. Part of what we trying to achieve was to introduce the dataset, provide a holistic perspective, and highlight what we see as the most salient results. This with the intention that other groups, and ourselves, would return to the results to examine certain aspects in greater detail. In retrospect, we agree with the reviewer that the main take away messages were lost in the original version, essentially because we tried to cover so much ground. We have taken the following steps to mitigate this.

(1) The text reporting our results has been revised to signpost the key questions and findings as explicitly as possible.

(2) Text has been added linking/signposting the sequence of analyses throughout the main article.

(3) A new figure 1 has been made, which signposts the questions addressed by the major sections of the results, alongside an annotated timeline of data collection.

(4) The discussion section has been expanded, with more of a focus on what the results mean from the perspective of informing policy.

(5) The PDF spreadsheets have been removed and replaced with tables and figures that are more easily parsed in the supplement document.

(6) The supplement has been reorganised substantially.

(7) We note, all data and models already were shared in an online repository. Code will be added to this at the point of acceptance for publication (i.e., when we know what the final post-review pipeline is).

It is important to note that adding this type of detail, to unpack the results, comes a cost to space. Nature Communications has a word limit of 5K words for the main text. For this reason, we decided to remove the free text analysis, and intend to focus exclusively on this in a further publication. The latter helps make the paper easier to follow and also helps us to meet the word limit requirements whilst simultaneously addressing the critiques. We have noted to the editor that we would be willing to add these analyses back in, but this would require permission to greatly exceed the word limit.

Regarding the request for a web application: this is a not an insubstantial software development job. However, we have coincidentally already been working on a web

app, this to align with a BBC1 Horizon Documentary on mental health that will air in December 2020. That documentary will report many of the results reported in this article. It also will serve as a call to action, where people can go to our website, complete the questionnaire. We intend that participants be provided with tailored advice based on the large corpus of lived experiences analysed in our LDA. Over subsequent weeks, we will assess the impact this has on peoples' mood and wellbeing. We note though, that this study and web app will not launch until later in the year. Given the current relevance of our findings here, we do not think that publication of this article should be contingent on this separate substantial body of additional work - especially if the free text analysis is reported in a separate article.

2. I evaluated the methods and statistical analyses to the best of my ability, but this was sometimes difficult because 1) not enough detail was provided, or 2) some of the statistical methods might need some explanation to be accessible to a general social science audience (I am writing from the perspective of a psychologist with substantial training in advanced quantitative methods but limited knowledge of LDA and machine learning techniques). Here are some points that I thought needed clarification:

We are grateful for these helpful suggestions and have addressed all of them within the main text. Our strategy in doing so has been to signpost key points and analyses within the main text, and due to space constraints, link out to more accessible and detailed statistical detail within the methods. Analysis scripts also will be made available at the point of acceptance for publication. This will be alongside models and data, which already were available online.

a) It is unclear whether any of the same participants were followed over time between the pre- and mid-pandemic waves, or whether these samples were completely independent. Were the 74,830 participants who responded to the PD-GIS-11 a subset of the 110,118 mid-pandemic participants? Were the 8,680 early-pandemic participants included in any of the analyses?

We have now made explicit in the methods that participants in the pre- and mid-pandemic phases were recruited using identical advertising methodologies – specifically, articles placed on the BBC2 Horizon site and the BBC news and homepage. The vast majority of traffic came from the latter two, where we were placed top and centre, making the study extremely visible to the general public. Data were collected once from each individual, with this coinciding with different time points, that is, rather than from the same individuals at different time points. We also have made clear that the early-pandemic participants were not included in the analyses, because they were not demographically matched (they represented a younger group), this being due to the study not being so prominent online during that phase.

Pre-pandemic and Peak COVID lockdown epochs have near perfectly matched demographics in terms of age, gender, handedness, employment status, earnings, ethnicity, first language etc. This also is now clarified in the main text and a supplemental figure showing this is included.

Finally, we clarify that the participants who responded to the PD-GIS were indeed a subset of those who completed the study mid-pandemic. This was an optional scale within the broader questionnaire instrument, completed in the same session. It was optional because the study was not advertised as being Covid-19 oriented, which avoids biased sampling, but also requires an opt-out for that section for ethical reasons, and to ensure those who completed it intended to do so accurately as opposed to skipping ahead to result. In terms of bias, we now include supplemental analyses showing that any biases in mental health measures for those opting in to the extended questionnaire are well below the cutoff for the negligible effect size range.

b) p. 5: "Differences in mood assessment scores were calculated for the densely sampled and closely matched Mid vs Pre-Pandemic epochs after factoring out age, sex, handedness, education level, first language, country of residence, occupational status and income". What is meant by "densely sampled" vs. "closely matched"? Does this mean that there was another version of the analyses that involved comparing demographically-matched samples? Or does it just mean that after factoring out the various controls, the two samples became closely matched? In any case, it would be helpful to see a supplemental table with statistical comparisons and effect sizes for the differences between the two samples. It is difficult to gauge possible differences from Figure 1e.

We now are more precise in our terminology. We refer to the two groups being near perfectly matched in terms of demographic characteristics (discussed in results and reported in detail in the supplement) - that is, without any requirement to adjust or factor out data. The samples were simply collected using similar promotion and at very large scale, via the same BBC websites, leading to a near perfectly matched demographic profile. Further details of the characteristics of the two samples have now been added to the supplement.

c)p. 5: I am confused by the rationale for calculating mood scores separately for each of the 31 days pre- and mid-pandemic, then running 31 comparisons, then reporting an average difference based on 31 comparisons. This seems to add unnecessary complexity—why not just compute an average mood score for the pre-pandemic epoch and post-pandemic epoch, and compare those?

Our rationale here was to demonstrate that the observed differences were not simply a consequence of some blip in the data, but instead, were consistent throughout the epoch. One can consider this also as a form of retest validation, whereby the same analyses are repeated, arriving at the same answer, with 31 independent data

samples. This seemed important because day-to-day fluctuations may be expected due to the potential impact of acute events such as media announcements by government and media. Nonetheless, we agree this level of detail complicated the results section. Therefore, we now report a statistical comparison of the data collapsed across the whole duration of the epochs as suggested.

d) Figure 2: What are those colorful squares in the right panel? This needs more explanation. I also don't understand why the X scale of the left panel is in SD units. What were the scores standardized with respect to? I strongly believe that it would more interpretable to know the level of agreement with each statement on the raw metric (i.e., x axis should be from 1 = strongly disagree to 5 = strongly agree).

We now have broken this information down across multiple figures. Specifically, Figure 2 now reports the % respondents on the frequency scale for each aspects of the mood self assessment questionnaire, side by side for the pre and mid pandemic epochs. This is then followed up with figure 3, reporting the statistical analyses, where that data is transformed onto a numerical scale with unit standard deviation, enabling statistical analysis with a GLM, that takes into account the various population factors of interest. (We note, this approach is necessary to enable potentially confounding factors to be accounted for whilst examining the interactions of interest).

Figure 3 now uses circle size to denote effect size. We also include the actual numerical data in the supplement. The reason for displaying this way is that it enables us to fit the many results into a single figure in a simple/not crowded manner. We could not bar charts with labels, there would be too many for one figure, and we already are at the figure limit.

As a general rule, when reworking the manuscript, we have added in sufficient detail in the results section for the reader to understand why we chose the approaches that we did.

e) p. 8: I am not familiar with canonical correlation analysis, and don't think the average social science reader would be either. Can you add a brief explanation of what this tells us? In Figure 3b, why are the last two rows empty? Does that mean that there was no correlation identified by the canonical correlation analysis?

CCA is one of the more established multivariate statistical methods for dealing with the issue of there being a great many possible bi-variate correlations between two sets of features by identifying latent variables that correlate across the two sets of features. This produces 'modes' of correlation capturing variance in multiple variables on both sides of the equation. It is particularly appropriate where there is substantial covariance within one or more of the matrices. This method has been available for some time, but it has become more popular again in recent years in the neuroimaging field. We now provide more description of this technique and why it

was appropriate in the paper, along with relevant reference. It also is the case that the CCA is now only used when interrelating the mood self assessment and PD-GIS measures. The trait vs PD-GIS analysis has been replaced with a more standard support vector machine approach, which we describe and reference in more detail. The reason for this is that it enables us to ascertain how much of each PD-GIS component variance (i.e., pertaining to self-perceived impact) can be explained by the traits in combination.

f) I did not understand what was going on in Figure 6, or the canonical correlation analysis that led to this. What are the two modes?

This figure and analysis has now been replaced.

g) Figure 7a would be more informative if it contained some information about the relative prevalence of the topics.

This has now been removed due to the free text data being covered in a separate future paper.

h) For the advice topic analyses, why were t statistics and p values reported instead of standardized effect sizes (both in text and in Figure 7b and the supplement)? This was strange, given the previous emphasis on effect size rather than p values. It makes it difficult to know how large the relative differences were.

Now removed due to the free text data being covered in a separate future paper as described above.

i) Basic details about LDA analyses were missing (and further details were not in fact given in Supplement 4). At the very least, how were data tokenized (e.g., only words, or also 2- or 3-gram phrases)? What were the α and β parameters? How were multiword collocations identified (if applicable)? How were topic scores calculated? Did the final sets of 5 negative topics, 10 positive topics, and 50 advice topics capture ALL topics that were identified by LDA, or were there some uninterpretable topics that were excluded?

Now removed due to the free text data being covered in a separate future paper as described above.

j) Most of the supplemental tables and graphs need much more explanation (e.g., details of statistical analyses, x and y axis labels, whether coefficients were standardized) than what is currently provided.

The supplement has been reorganised into a more linear format that we trust is now easier to interpret.

3. Based only on the Kaiser rule of including components with eigenvalues > 1 (which has some problems, see <https://stats.stackexchange.com/questions/253535/the-advantages-and-disadvantages-of-using-kaiser-rule-to-select-the-number-of-pr>), the authors extracted 11 components from the pandemic impacts questionnaire. Given that so many of the subsequent analyses involve these resulting component scores, I would have expected to see a more thorough exploration of the dimensionality of this questionnaire, using different rules for determining the number of components to extract (e.g., visual inspection of the Scree plot, MAP test, parallel analysis, very simple structure criterion), as well as a consideration of the face-validity/interpretability of the resulting components.

There is no one correct method for determining the number of factors to retain in a PCA, but we completely respect the importance of ensuring the utilised approach is rigorous. We agree that it is an important issue in the case of the PD-GIS, as a key question that we ask is whether the way in which people perceive the pandemic to have affected them is high dimensional, that is, as opposed to being explainable on a monolithic positive-negative scale. Consequently, in the revised manuscript, we apply an established permutation-based approach to estimate how many components we have statistical evidence for. This would suggest high certainty of at least 7 components. The Kaiser convention indicated 11. Taking the elbow of the scree is prone to differences in interpretation, especially here, where there are multiple such elbows (evident in the scree plots). Based on the above, we have decided to refocus the analyses on the 7 component decomposition of the PD-GIS.

To be fair, most of the components reported in Figure 2 seem to make sense, but there are some exceptions. For example, a) "Grieving" doesn't seem to fit with the "Concerned with health" dimension, b) might "More in touch with loved ones" belong in the "More time for people" or "More time, less stressed/tired" component?, c) is the isolated & disrupted component coherent (or would they perhaps be better summarized by two distinct components)?, and d) does "appreciating simple things in life more" really belong in "Improved environment"?. In any case, it would be helpful to know how consistent the number and interpretability of the components are based on different criteria.

We now provide analyses of a somewhat less granular 7 component breakdown.

It would also be nice to see some discussion of how these 11 dimensions conceptually map (or don't) onto the 5 negative impact and 10 positive impact topics identified via LDA. Also, I could not find Supplements 4b and 4c (mentioned at the bottom of p. 13).

Now deleted per above comments.

Relatedly, I request that the authors report the full text of the Pandemic General Impact Scale in the supplement. This is described in Supplement 1, but it would be helpful to see the exact wording that participants saw. For example, it is unclear what “improved natural environment” means.

This has now been included in the supplement as requested.

Minor points

4. Data and analysis scripts were not available. For transparency, I encourage the authors to consider uploading the data and analysis scripts required to reproduce the results reported in this paper to an online repository such as the Open Science Framework. The reporting summary states that data are freely available, but where?

Data and models already were shared online via the UK data service and we apologize that this was not clear in our original submission. We have not shared code yet. This is because we share the code for the final analyses post review, that is, when any reviewer-led adjustments/additions to the analyses have been incorporated into the code. We confirm that code will be shared though and wholeheartedly agree that this is key for transparency.

5. In the introduction, it is stated that “addressing these knowledge gaps during the pandemic provides a unique opportunity to understand the nature and psychological basis of mental health resilience and vulnerability in the general population” (p. 2). I was hoping to see greater discussion of how the current findings or the general methodological approach can contribute to knowledge about mental health beyond a pandemic context. For example, some more general lessons might be that 1) the effects of any given stressor/life event might have differential impacts depending on sociodemographic variables, context, and personality traits, and 2) topic models of open-ended text responses could be used to identify promising personalized interventions more generally (not just for coping with a pandemic)—to me, this seems like one of the most exciting contributions of this paper (which is why I suggested making these data more easily-interpretable in point 1 above). The discussion would be improved with greater contextualization of the results within the broader literatures on personalized mental health interventions and the use of NLP for understanding mental health.

We agree that the results have broader relevance beyond the pandemic, and have expanded on these points of broader relevance in the discussion. As noted, the free text aspect has been removed, to enable the current manuscript to be easier to follow, and given that the free text analysis warrants a publication in its own right, on reflection based on the reviewers’ feedback in the previous round.

6. Here are some minor points of clarification/unclear sentences/typos I detected:

a) p. 2: What is meant by a “many-to-many mapping relationship”?

This term has been adjusted and clarified. What is meant by this, is situations where there are multivariate relationships between sets of variables. A good example is in neural coding, where different combinations of the same neuronal population can code for many more unique items than if they coded for those items in a 1-1 manner.

b) p. 3: *“This resulted in 215,886 Pre-pandemic, 8,680 Early-pandemic and 110,118 Mid-pandemic datasets in the reported analyses”.* It seems weird to refer to participants as “datasets”.

Now corrected.

c) Figure 1 caption: *“Sociodemographically *mediated* differences” implies that the differences in population mood scores from pre- to mid-pandemic are *because* of differences in the sociodemographic composition of the two samples. “Sociodemographic correlates” would be more accurate.*

Now corrected – we are careful to use the terms covary, relate, or correlate throughout so as not to infer causality.

d) Figure 1c: *needs a key or note in the figure caption to specify which colors correspond to pre- and post-pandemic.*

Now added.

e) p. 5: *“When analysing large-scale data, very small effects will tend to have highly significant statistical values. Therefore, the better measure of significance is effect size.”* The difference uses of “significance” is confusing here. I would change the second “significance” to “practical importance”.

Now corrected.

f) p. 5: *“Increases in depression was lower for younger adults and higher for older adults (-0.2SDs to 0.1SDs).”* This sentence is unclear: did younger adults decrease in depression?

Correct- small decrease for younger adults, now clarified in the text.

g) Figure 5a-b: *“Change in” implies change between pre- and mid-pandemic, but the impact scale was only administered at one time point. “Change in” should be deleted.*

We have now updated the legend to make clear what is meant by ‘change in’ – i.e. this is the difference in the average reported scores between the two samples.

h) Figure 3a: *y axis needs labels or numbers—it’s difficult to cross-reference.*

We are not sure what is being referred to here as this is a cross correlation matrix. We have modified legends and labelling to clarify, but are open to other suggestions.

i) Figure 3b: p cannot be .0000. Change these to $p < .0001$.

Now corrected.

j) p. 12: “mediated along” should be “correlated with”. “mediation” implies a process model with X explaining Y via mediator M .

Now corrected.

k) p. 13: “What has been most POSITIVE about the lockdown” was repeated twice. The second occurrence was supposed to be the question about coping.

Removed.

l) p. 16: Speaking as a personality psychologist, extraversion should definitely be spelled with an a (extraversion), not extroversion. This is the scientifically accepted spelling. See <https://blogs.scientificamerican.com/beautiful-minds/the-difference-between-extraversion-and-extroversion/> for a deep dive on the origins of the misspelling.

Thank you – now fixed, and we enjoyed reading about the origin of the misspelling.

m) Page 3 states that people who completed the questionnaire < 6 minutes were excluded, whereas page 18 defines “unfeasibly fast” as < 4 minutes. Which is correct?

This typo has now been corrected. A 4 minute cut-off was applied (based on our timing people completing the questionnaire). This criterion was decided prior to analysis by consensus amongst the study team.

Response to Reviewer #2

This manuscript reports a study based on a large-scale predominantly national survey. The topic is clearly timely and the study has been conducted according to advanced and methodologically sound approaches, including a sophisticated analytic plan. The reporting of the study is generally good and clear.

I do not have any major concerns, only a few suggestions:

We thank the reviewer for their positive comments and for noting the study was generally good and clear in terms of reporting.

Abstract: This is not fully clear to me; could this be rephrased?

We agree - the abstract now has been replaced with one that we feel has more of a linear style, that is, to more clearly specify the purpose of the study approach and implications.

There are currently a number of ongoing Covid-19 related surveys in the UK, aimed at assessing the mental/physical impact of the pandemic. The author may want to cite them (or at list the main ones) highlighting the strengths/weaknesses of their sample in relation to these existing ones.

We have expanded our referencing substantially and the limitations section, to account for these suggestions.

The discussion should highlight the limitations of convenience samples (<https://pubmed.ncbi.nlm.nih.gov/32502467/>)

Discussion of population sampling approaches now included in both introduction and discussion sections of the paper. This includes discussion of important challenges for achieving representative samples. In particular, we note the following key points. (1) The study has a highly (unusually) inclusive sample across age, ethnic, clinical and demographic variables (e.g. relative to Biobank, which is now cited with percentage). (2) such large scale data allows us to model many relevant population variables. (3) The study is unbiased insofar as it was not advertised as having a focus on Covid-19. (4) those who completed the PD-GIS were very similar to those who did not (indicating a lack of completion bias for this crucial instrument deployed at the peak of the initial lockdown).

Response to Reviewer #3

This paper presents data-driven analyses from a unique and incredibly rich dataset that, notably, includes mental health measures pre- and post-pandemic! Given the unprecedented changes to life during Covid-19, I thought the decision to adopt a data-driven approach was warranted. However, I also felt that the authors had decided to report a bit too much to yield a coherent report. There is nothing wrong, of course, with reporting various analyses with different outcome measures, but this approach seemed to have resulted in insufficient detail of many of the specific analyses, so much so as to prevent readers from understanding what the analyses mean. It does not help that the authors go from analyses to analyses and from outcome to outcome with very little integration and explanation of how each piece fits together. The omission of important information varies from the cosmetic (e.g., failing to indicate what each color signifies in figures) to the fundamental (failing to specify the design: between or within-subjects). Thus, I am keeping my review fairly short, only highlighting some of the issues with the presentation and the analyses of this incredible dataset. That said, I do hope to see a revised version that is more detailed, organized, and ready for the limelight!

We thank the reviewer for noting the uniqueness and richness of the dataset, which includes data pre-pandemic and peak lockdown.

To clarify what we sought to achieve, we felt that it was important to provide a complete and holistic analysis of the data, rather than “salami” slice the results into numerous papers not able to control for the relevant confounders, as seems to be the *mode du jour*. This is because a main point we are making is that the way the pandemic has affected people is highly idiosyncratic, but also statistically predictable based on a multivariate combination of demographics and circumstances. Also, given the relationship of impact to many variables, a key point also is that it is important to include such variables in order to disambiguate observed relationships. In practice, with data of such scale and scope, this presents some major challenges when reporting the results within a fixed space. Our intention in a way was to introduce this unique dataset, provide a holistic view that enables the relative and combined importance of different population factors in predicting mental health measures and self perceived impact, highlight the most salient such factors, and make the data available for more detailed scrutiny.

In retrospect, we entirely agree that in pursuing the above aim, our original version lacked the structure, narrative and detail needed to make it accessible for the reader.

In response to this, we have substantially updated our results and discussion sections to provide more detail regarding each analysis and its interpretation; as well as to make the paper considerably more organised. As per our responses to other reviewers (who felt the same way), our strategy for dealing with this challenge has been as follows.

(1) The text reporting our results has been revised to signpost the key questions and findings as explicitly as possible.

(2) Text has been added linking/signposting the sequence of analyses throughout the main article.

(3) A new figure 1 has been made, which signposts the questions addressed by the major sections of the results, alongside an annotated timeline of data collection.

(4) The discussion section has been expanded, with more of a focus on what the results mean from the perspective of informing policy.

(5) The PDF spreadsheets have been removed and replaced with tables and figures that are more easily parsed in the supplement document.

(6) The supplement has been reorganised substantially.

(7) We note, all data and models already were shared in an online repository, but this was not sufficiently clear in our original submission - our apologies. Code will be added to this at the point of acceptance for publication (i.e., when we know what the final post-review pipeline is).

We note that adding this type of detail has come at cost to space. Nature Communications has a word limit of 5K words for the main text. For this reason, we decided to remove the free text analysis, and intend to focus exclusively on this in a further publication. The latter helps make the paper easier to follow and also helps us to meet the word limit requirements whilst simultaneously addressing the critiques. We have noted to the editor that we would be willing to add these analyses back in, but this would require permission to greatly exceed the word limit.

Also, findings are now integrated with more of a narrative style in the discussion, where we also added a section on wider implications (beyond COVID).

We hope you agree that these changes have also aided in making the paper easier to read and interpret.

1. Many of the graphs are difficult to interpret due to insufficient information provided in the legends. For example, Figure 1c presents anxiety over 31 days pre and post-pandemic. OK, but which color indicates which period? Is the reader supposed to infer those visually from Figure 1a? But if that is so, why are the same colors (blue and red) used to signify a completely different thing in Figure 1e (increase or decrease)? And again, the reader should not be left to infer that blue means a decrease, and red means an increase. The figures and their legends should be interpretable without having to read the entire paper. In Figure 4c, is White the comparison category? Also, why are d and e reported in the legend but not shown in the figure? Obviously, these are cosmetic issues, but they are also pervasive. As a

reviewer, it is not my job to be a copy-editor, so I will not go on in detail about all the issues with the figures here, but the feedback is clear: Make your figures interpretable.

We apologise that the figures lacked sufficient legend information for them to be interpreted sufficiently. Making figures for such complex data is a particular challenge. We now have reworked, and in many cases completely replaced the figures with alternatives that we believe should be simpler for the reader to interpret.

2. Though never explicitly specified, I am assuming that the authors are using a between-subjects design, meaning they are comparing different groups pre and post-pandemic. That said, some of the writing seems to suggest within-subjects design: "Six mental health measures were calculated and compared separately for people who were sampled at each of 31 days post-launch, effectively providing 31 independent replication analyses." If the analyses are indeed between-subjects, then these are not really pre–post analyses, which typically implies a repeated, longitudinal design. We then must consider whether the participants in the two groups were randomly assigned to those groups, which is not the case. Finally, we must consider how those two samples might be different from each other (which might account for any differences that are observed). And the list of such factors is likely long: from the way participants were recruited to self-selection. For example, people who were more anxious to begin with might have been more likely to participate in a survey about COVID. Statistically controlling for a few factors cannot isolate COVID-19 as the direct causal factor. Yet, the analyses are interpreted exactly as a result of COVID.

The reviewer is correct about the design, and this has now been further emphasised in the updated results, methods, and in the abstract. The two samples (pre-pandemic and mid-pandemic) were practically identical to each other in terms of multiple key demographic measures. Early-pandemic, this was not the case with reference to age specifically. Therefore, we excluded the smaller N early-pandemic data from the analyses. The reason that the datasets are so similar is that the study was advertised in the same way and place. That is, as an article placed at the top and centre of the BBC homepage and BBC news homepage. It literally is not possible to buy advertising space of such prominence in the public eye. We use Google Analytics to track where people come from on our site. As expected, the vast majority transitioned into our landing age from these same locations, advertised in the same place and same way. This fact is now stated, and graphs showing the comparisons are clearly signposted and included in the supplement. Despite this, as noted we also controlled for key variables across all analyses. We believe the extent of our statistical control is unusually rigorous amongst mental health research and also that the large sample size allows such approaches to be more than amply powered (we can in fact detect with high certainty effects that are of negligible scale, or that relate to small sub-populations within the UK). Random assignment of

participants to each sample was not possible due to the unpredicted nature of the pandemic. However, our sample is more representative and inclusive of the UK population than randomly sampled studies of established cohorts - this by dint of the unique reach of the BBC. We also do not suffer the same sort of biases that established cohorts have, that is, in terms of who it is that is most likely to engage in repeat or additional assessment. In the methods, we now make clear that the recruitment approaches for the two samples were identical, and that the recruitment materials did not make mention of COVID-19, thereby minimising the risk of COVID-related selection bias. Furthermore, we provide analysis of the (negligible scaled) biases within this very large dataset insofar as they pertain to the mental health measures of those who opted to complete the detailed PD-GIS scale.

3. I did find the moderation analyses by demographic factors to be most informative and interesting (Figure 1e). I wish the authors had explored these changes in anxiety, depression, and other outcomes measured with validated scales more in terms of other factors, such as current living situation, personality, and preexisting mental disorders, more—the same way they explored it with for the perceived impact outcome. I also found it confusing that the authors talk about demographically “mediated” effects. Mediation implies that X causes M, which in turn causes Y. This is not the case here as COVID-19 does not cause somebody to be man or woman, for example. These are moderation analyses, whereby the effect of COVID on Y depends on various demographic factors, M (the moderators).

The point regarding moderator vs mediator is well made – we have corrected this in the main text. We also have expanded the analysis of moderators of the established mental health measures as requested insofar as the data that we have space to allow for this. We also are making the dataset available for researchers more broadly. It seems reasonable to suggest that others could focus in much greater detail on the mental health questionnaire specifically, including with the potential that our intended further data collection will enable as events unfold. This we feel is beyond the scope of the current article.

4. I did find the analyses with the perceived impact to be interesting and informative. But while the authors acknowledge early on that these variables are capturing perceived impact, this is quickly lost in the deluge of analyses and by Figure 5, the authors proudly announce in the figure caption that they are showing “Impact” of Covid-19 as a function of mental disorder. Though this may seem like another cosmetic suggestion, it is a serious concern. The authors are effectively claiming—with little evidence—that they have studied the impact of COVID-19, when all they are showing is what people, in completely retrospective reflections, perceive as factors that have impacted them during COVID. The limitations of retrospective perceptions in making conclusions about actual impact are well documented and I will not elaborate further.

We completely agree with the reviewer's point about being careful to be clear this is *self-perceived* impact of the pandemic; we have updated the paper throughout to ensure we refer to "self-perceived impact" (or equivalent terms) rather than "impact". We also have included analyses interrelating self perceived impact with the dimensions that are shown to differ pre and post pandemic. This provides further evidence that population characteristics that relate to perceived impact, also relate to observed differences, albeit cross sectionally at this time. We have chosen our words carefully in that analysis to make clear that it is not the same as a longitudinal analysis.

5. The discussion is another missed opportunity to integrate the findings and outline the limitations of the design and the analyses.

The discussion has been substantially updated to integrate the findings, and clarify limitations in more detail, as well as considerably enhancing referencing to the extant literature.

6. The authors casually state that "as a quality control people who completed the questionnaire too rapidly (defined as under 6 minutes) were excluded." This seems quite arbitrary. When was this decision made? Was this decision preregistered? Was it at least made before looking at any of the analyses? How did the authors decide on 6 minutes?

This decision (in fact 4 minutes - 6 was a typo!) was made by the lead investigators prior to data analysis taking place. It was based on the approximate minimum observed time taken to complete the survey if clicking through as rapidly as possible while reading the questions. We concur such a threshold will always be arbitrary but feel it is better to have a concrete pre-decided threshold rather than include data that by definition would have been collected from subjects who did not read the questions. We feel that the rationale is similar to the sort of performance cutoff filter that is often applied in objective cognitive assessment (i.e., performance that is recorded below the feasible range).

In sum, while data-driven analyses of such rich data have enormous potential to inform our understanding of who is most impacted by Covid-19, adopting a data-driven approach does not mean quickly reporting a multitude of analyses without sufficient detail and little integration or interpretation. And even the fanciest analyses and richest datasets can still not make up for careful, thoughtful interpretation of what the results mean and what they do not.

We have taken the reviewer's critique on board, substantially amending the manuscript and supplement throughout. We hope that they agree the manuscript is now considerably clearer.

Response to Reviewer #4

This review is for "Dimensions and modulators of behavioural and mental- health change during the Covid-19 pandemic" (NCOMMS-20-28403-T). Overall, there are several things to like about this manuscript. It is generally well-written and timely, and both factors work to its advantage. Ultimately, my main critiques around this paper are really just centered in on the question of "what is the content, scale, and scope of this work's contribution to the literature?" I found my main question somewhat difficult to answer. Much of my struggles here revolved around the fact that I didn't feel like there was really enough information to evaluate everything that was going on here. In many ways, and in many places, the manuscript feels a bit like a "rush to the finish line", so to speak. For many of the analyses, I found that the statistical reporting was rather incomplete, with a lot of the reporting feeling like there's an implicit statement of "please just trust that we were thoughtful and rigorous in our analyses". I don't doubt that the authors were, but given the size, source, and overall nature of the data, I would have expected perhaps more nuance and clarity about everything that's going on here. For things like the PCA and LDA, I didn't really see any evidence for the statistical decisions made, which is particularly necessary given their unsupervised nature (i.e., one can arbitrarily apply numerous criteria to determine the number of components, topics, etc.). Additionally, there simply wasn't enough information to really unpack these intermediate steps, let alone the results taken on the data.

We thank the reviewer for these thoughtful comments. Fitting an article of such scale and scope within the set word limit has been a massive challenge. We agree though (as evidently do the other reviewers) that more detail was required. The paper has now been substantially reworked to provide more methodological information and rationale throughout, so that the reader can understand exactly what was done, and our decision making process. As per above response - our approach in redrafting has been as follows.

- (1) The text reporting our results has been revised to signpost the key questions and findings as explicitly as possible.
- (2) Text has been added linking/signposting the sequence of analyses throughout the main article.
- (3) A new figure 1 has been made, which signposts the questions addressed by the major sections of the results, alongside an annotated timeline of data collection.
- (4) The discussion section has been expanded, with more of a focus on what the results mean from the perspective of informing policy.
- (5) The PDF spreadsheets have been removed and replaced with tables and figures that are more easily parsed in the supplement document.

(6) The supplement has been reorganised substantially.

(7) We note, all data and models already were shared in an online repository, but we did not make this sufficiently clear in our original submission. Code will be added to this at the point of acceptance for publication (i.e., when we know what the final post-review pipeline is).

In order to make space for this, we have removed the free text analyses. We have indicated to the editor that if allowed more space, we can add those analyses back in. Our intention though, is to publish those analyses elsewhere, where we will have sufficient space to explain the methods and discuss implications.

Given the stated goals of the paper, I was somewhat surprised by the fact that, for the most part, the presentation of the data and results leaned towards the cursory and descriptive side of things, without really unpacking the meaning and nature of the lockdown specifically on the participants' psychology. Much of what was reported here was not particularly surprising (not that the findings need to be, or should be, in order to be reported/published). But, I felt that this work was fairly disconnected from just about every relevant literature there is. Out of 20 references, at least 50% were specific to COVID/coronavirus. However, the health psych literature broadly defined, large swaths of the social psych literature, the and massive amounts of psychopathology literature are relevant, and based on everything that we know from each of these rich fields of study, I don't think that there is anything being reported here that wouldn't be predicted and hasn't been shown, to some degree or another, in smaller samples (in many cases, still very large samples, just not as large as this one). Even the general psychometrics literature would probably chip away at much of measures here, citing problems of shared method variance, the extensive reliance of self-reports, and so on.

We agree and, as per our above response, we have greatly increased detail, enabled by removing the free text analyses to make more space. We also agree that there are rich fields of study outside of the COVID context and that it is helpful to relate the findings to that work more explicitly. We hope the reviewer will appreciate that we still are limited in the extent to which we can discuss non-COVID literature due to the length restrictions of the manuscript. Nonetheless, the discussion has now been expanded to better integrate relevant research. We also substantially expand our discussion of the implications, both in the context of the pandemic, but also beyond, taking into account both the negatives, but also prominent positive aspects of self-perceived impact.

We respectfully note that the other three reviewers all considered there to be substantial novelty and value in this work. Reviewer 1 noted the 'precise estimation of effect sizes and meaningful investigation of differences between subgroups'

'comprehensive set of predictors' and focus on 'differential impacts of the pandemic on different groups (rather than focusing on average trends). They stated 'the paper contributes timely insights' 'to inform mental health interventions during these times, and 'outside of a pandemic context'. Reviewer 2 considered the study was 'timely' and 'has been conducted according to advanced and methodologically sounds approaches, including a sophisticated analytic plan'. Reviewer 3 noted that this is 'a unique and incredibly rich dataset that, notably, includes mental health measures pre- and post-pandemic'. However, we concede that the way in which the article was originally reported failed to emphasise the key findings and novel aspects. We feel that we have dealt with this issue thoroughly in the revised manuscript. In particular we highlight the following novel aspects.

On a meta level, one of the key strengths of our holistic approach, with a very large dataset, is that we can gain insights into the many moderators of pandemic impact on multiple dimensions of mental health and behaviour, both positive and negative, whilst robustly controlling for each of them. This provides novel insights regarding the idiosyncratic ways people have been affected. For comparison, other Covid-19 studies typically focused on 1-2 mental health symptom domains in isolation; in many cases, Covid-19 cohort data have been sliced into separate papers, none of which control for confounds recorded within the study and reported separately elsewhere. A product of this, beyond thorough controlling amongst correlated variables, is that we are able to ascertain *the relative importance* of population variables, which is not possible when they are examined in isolation. Here, for example, we can see that age, home context and work arrangements have what are often very large relationships to the self-perceived impact of the pandemic. Traits and preexisting conditions have small to medium sized relationships. Many other variables have statistically significant relationships, but in the grander scheme of things, focusing on the above seems sensible.

Other individual novel aspects include:

Using a citizen science approach that does not include mention of C19 in the advertisement materials, thereby avoiding biasing results.

Demonstrating that, counter to reports from some smaller scale studies, some measures of mood and insomnia appear on average to be improved during the lockdown when the population is sampled in this very large scale and inclusive manner, and that correspondingly, many people report that their perception is of being better off than before in some respects.

Analysis of the relevance of compulsivity traits to self-perceived pandemic impact - this enabled by use of our detailed compulsivity scale. This was not previously possible as trans-diagnostic measures have only recently been validated.

Demonstration that environmental factors such as pleasant outside space and personality factors such as compassion and conscientiousness can confer benefits in terms not only of mental health measures, but also resilience against negative consequences of events such as a pandemic.

The finding that there is much prominence in positive perceptions of the impact that the pandemic lockdown has had on daily life and wellbeing, but that variability in such positive sentiment is best described across multiple dimensions that can be predicted based on population factors.

We also agree with the view that confirmation of expected findings at scale in a manner that allows for precision of estimates and controls for many other variables has a not inconsiderable scientific value.

We note that the other reviewers highlighted the richness and uniqueness of the dataset and importance of the findings. We certainly agree though that understanding of the Covid-19 situation needs to be informed by studies using a variety of methodologies, not only self-report, and have added this to the end of the discussion section accordingly. But we also believe that much can be learnt from peoples' expert lived experiences and agree that the findings have relevance beyond the pandemic. For example, the prominent relationships of positive aspects of perceived impact with reduced commuting and access to pleasant outside space in such a large population sample can help inform pragmatic/implementable changes that affect positive change in the longer term, as well as augment resilience to challenges like pandemics.

All of this said, I think that there remains plenty of value to this work, but I'm not sure if it really is coming to the surface or shining in the ways that it could. I would ask that the authors provide a vastly more detailed and extensive set of supplementary materials / data availability / code availability than what I found for this work (unless I'm missing something critical) so that reviewers could really dig in and do a deeper evaluation of the work that has been done here. I would also ask that the authors perhaps revisit their framing of the work being done here and think about what boundaries might be placed on the findings giving some of the not-insignificant psychometric considerations of the data as-is, and as it has been collected. And, as noted above, I would love to see this work much more deeply connected with the relevant literatures, linking up to relevant findings that have been discussed elsewhere (particularly in health psych, but much beyond this as well), and discussing/demonstrating how this work builds on past research and brings new things to light that we might not already understand about factors such as individual differences, social media, relationships, demography, and health outcomes

We are grateful for these helpful comments and have now substantially revised the article end to end to address them, providing much greater detail on the analyses

decisions, results, discussing implications, and integrating more broadly with the literature including substantially broader referencing.

REVIEWER COMMENTS

Reviewer #1 (Remarks to the Author):

This is a review for manuscript #NCOMMS-20-28403A-Z. Compared to the previous version, this version is much more focused and readable, with effective use of figures. I commend the authors on these improvements, which will greatly facilitate communication of these rich and large-scale data. However, there are still a quite a few substantive issues that should be addressed, most of them pertaining to the interpretation of the PD-GIS.

1. The two items with the largest component scores for “More time for people” are those labelled as “Happier as more time for people at home” and “Better connected”. I appreciate the full reporting of the PD-GIS item text. This revealed that “Better connected” is shorthand for “I am better connected now with the people I live with”. This short label should be updated to reflect this (e.g., “Better connected with cohabitants”). In addition, the component would be more appropriately interpreted in terms of connectedness with cohabitants. If so, these component scores are confounded with cohabitancy status, because people who live alone would be unable to agree with these statements. Thus, the effects of the demographic and trait predictors of this component may be entirely confounded or at least overestimated.

For example, if older people are more likely to live alone, this would explain why older people have lower scores on this component (rather than actual changes in relationship quality). Similarly, if “time with people” really means “time with housemates”, it may be tautological to say that “People living alone score...lowest for more time with people” (p. 14 in the PDF). Another example of a potentially inaccurate sentence: “people with psychiatric and neurologic conditions were less likely to report increased connectedness during the pandemic” (p. 22). One way to deal with this would be to control for whether participants are living alone, or perhaps exclude such participants from analyses involving this component.

2. I would like to see more context on how the PD-GIS was developed. Who came up with the items? What were the criteria for writing and retaining items? Did you initially generate a larger pool of items, and if so, how did you end up with the final set of items?

Note that the editor asked whether the removal of the free text analysis is detrimental to the impact of the work. Although I thought that the free text analysis was one of the most intriguing aspects of the previous manuscript, I also felt that the previous manuscript was trying to do too much (at least for a short report). The revised manuscript already contains more than enough data without the free text analysis. I hope to see the free text analysis of the advice topics reported as a separate paper.

However, the authors might consider including the free text analysis of the positive and negative impacts in the supplement to provide further context on the PD-GIS. Doing so would demonstrate the extent to which the PD-GIS comprehensively captured the positive and negative impacts that

participants spontaneously generated (e.g., Did the PD-GIS leave out any frequently-mentioned impacts? As mentioned in my previous review, how do the PD-GIS dimensions conceptually map onto the topics identified via LDA?).

3. Some of the questions in the PD-GIS were not worded as increases or decreases, and therefore cannot be interpreted as “impact” of Covid-19. For example, the two questions about health concerns (“I am concerned about my personal health”; “I am concerned about the health of my loved ones”) were not worded as being “more” concerned. In the absence of a comparison against a pre-Covid-19 baseline, differences between groups cannot be interpreted as differences in increased health concerns due to Covid-19 (e.g., “increased health concerns scores higher for people with pre-existing concerns...but were greater still for people with anxiety disorders and OCD”, p. 16). Instead, such differences may reflect general differences in health concerns irrespective of the Covid-19 context.

Similarly, the “positive outlook” questions are not about the impact of Covid-19 on the participants’ lives, but about the participants’ sense of optimism about the impact of Covid-19 on the economy and society as a whole. In some places, the wording implies a *change* in positive outlook (e.g., “those of middle working age – who reported...reduced positive outlook”).

To ensure accuracy, the authors should rewrite sentences like the above in terms of group differences rather than impact (e.g., “People with pre-existing conditions had higher health concerns compared to people who did not have pre-existing conditions”). The authors should also do more to discuss the limitations of measuring impact using retrospective self-reports (compared to a longitudinal design). This was a concern raised by a previous reviewer that was not sufficiently addressed.

4. In the previous version of the manuscript, the authors identified 11 underlying dimensions. In this version, there are 7 underlying dimensions. Without seeing the author’s response to the reviews, it is not clear to me how the analyses changed to produce this rather different summary of the PD-GIS.

5. I am wondering why the authors chose to use component scores rather than to create scales based on the highest-loading items for each component. I see two major downsides to using component scores: a) because a PCA is likely to produce different solutions in different populations, other studies that use the PD-GIS are unlikely to have the same weights or even the same number of components, making comparisons between studies difficult, and b) the components cannot be cleanly interpreted in terms of a few conceptually related items (instead, they capture a blend of all of the impact items, just to varying extents).

Relatedly, in the Method (p. 25), the authors state “Where appropriate, scales were summarised in the following steps...subscales estimated using principal component analysis”. Does this mean that all scales were estimated using PCA? For example, it was unclear to me whether PCA was used only to produce a six-factor summary of the Big Five measure, and if scale means were then used (e.g., if “Introverted” only included “Talkative”, “reserved”, “tends to be quiet”), or whether component scores that included all 18 items were used. The latter approach would be extremely unconventional.

6. p. 19 states that self-security and compassion were associated with reduced scores on disrupted lifestyle and health concerns, but Figure 10 seems to show that only self-security is associated with reduced scores (and compassion is increased with increased scores, but there is no indication of significance in the figure). This is mentioned again in the discussion (“positive impact with compassion”, p. 22), but does not seem to be true. That sentence in the discussion also mentioned extraversion being associated with positive impact, but again, extraversion seems to be negligibly associated with the impact dimensions in Figure 10.

7. The Figure 10 caption should note whether all correlations were statistically significant (it is unclear whether “all $p < 0.0001$ ” refers to each of the correlations, or only to the multiple-predictor models). Also, “light green” should be “light grey”.

8. It was unclear which specific result this sentence was referring to, and whether this claim can be substantiated: “Our results indicate an important reason for this observation [about clinically vulnerable people reporting a worsening in their mental health] is likely to be isolation and disconnectedness” (p. 21).

9. The abstract states that these data are “objective”, but these data are “subjective” in the sense that subjective self-reports were used.

10. “the impact of the pandemic on mental health is likely to be mediated by a variety of factors” (p. 3) – “mediated” should be “moderated” or “modulated”

11. In Figure 3, the colors need a key (which color corresponds to which direction?). Also, the caption states “Younger adults were less depressed whereas older adults were more depressed”, but this misleadingly implies that younger adults had lower depression scores compared to older adults (whereas the finding is about a relative change from the pre-pandemic baseline).

12. In the supplement, Section 7, what are the estimates? Are they standardized regression coefficients? If so, this should be mentioned.

Reviewer: Jessie Sun

Reviewer #2 (Remarks to the Author):

I think the current version is much improved and reads very well. I do not have any other comment

Reviewer #3 (Remarks to the Author):

I congratulate the authors for their thorough and thoughtful revision of the paper. My primary concerns have been addressed.

I still feel that the paper presents a bit too much information in the main manuscript. More specifically, a large portion of the paper is focused on presenting predictors of the self-perceived reports of pandemic impact. I do not believe that these retrospective analyses deserve as much prime journal space, especially given that the main contributions of the paper are the comparisons between mental health before and after the pandemic.

I also wondered whether the authors might want to more prominently present comparisons or trends over time. In particular, what is currently missing is the initial impacts of the pandemic (e.g., in March, early April). In another manuscript that I just reviewed with different large representative data from the UK, well-being suffered most in the first two weeks after the first case of the virus reached the UK. After the initial shock, well-being gradually returned to normal by June. Beyond these specific findings that I cannot cite for obvious reasons, the key question is whether the relatively minor differences in mental health observed in the two epochs in the current paper might be due to an adaptation effect, whereby people recover after an initial drop in mental health.

Finally, a super minor comment to consider replacing "vs." with "by" in the titles of many of the figures.

Thank you!

Kostadin Kushlev

Reviewer #4 (Remarks to the Author):

The authors have done a very nice job with their revisions. The manuscript's presentation and clarity are a huge step up from the previous version, and it is my perspective that they have put in a lot of work into making this a nice, linear manuscript with a clear purpose and goal, and with the analyses being very well-documented and described. In these regards, I think that this is a highly successful revision. The manuscript is now much more well-grounded in several relevant literatures, and I again appreciate the thoughtfulness that went into this revision.

The only real lingering critique that I would have to offer is still a bit of a hold-over from my initial review: the ultimate presentation of this work feels as though it's a cursory "description" of phenomena that we already expected, anticipated, and have been shown elsewhere in various forms. I will reiterate that there is certainly value in this — however, it leaves me wanting a deeper/more extensive interpretation of the patterns. The correlations themselves are intuitive and gel with what we'd expect, and there aren't any earth-shattering revelations: specific psychodemographic subpopulations who are already at-risk for in any stressful/taxing scenario (short and long term alike) are experiencing those issues in a more pronounced fashion during covid. I would nudge the authors to go further than they are

currently going: there is already a sense of what we (i.e., policymakers) *should* be doing with this information. What does this information suggest we should do that might not have been suggested by past work? For example, does this information help us to identify subpopulations that might benefit from post-hoc assistance? If so, what forms of assistance? And so on.

RE: Dimensions and modulators of behavioural and mental-health change during the Covid-19 pandemic. Ref NCOMMS-20-28403-T

We are grateful for the reviewers encouraging and helpful comments. All points raised have been addressed below and the article has been substantially updated. We feel that it is greatly strengthened and hope that you will now consider it ready for publication.

Adam Hampshire (on behalf of the authors)

Reviewer #1

This is a review for manuscript #NCOMMS-20-28403A-Z. Compared to the previous version, this version is much more focused and readable, with effective use of figures. I commend the authors on these improvements, which will greatly facilitate communication of these rich and large-scale data. However, there are still a quite a few substantive issues that should be addressed, most of them pertaining to the interpretation of the PD-GIS.

We are grateful for the helpful further comments, which have been addressed point by point below.

1. The two items with the largest component scores for “More time for people” are those labelled as “Happier as more time for people at home” and “Better connected”. I appreciate the full reporting of the PD-GIS item text. This revealed that “Better connected” is shorthand for “I am better connected now with the people I live with”. This short label should be updated to reflect this (e.g., “Better connected with cohabitants”). In addition, the component scores are confounded with cohabitation status, because people who live alone would be unable to agree with these statements. Thus, the effects of the demographic and trait predictors of this component may be entirely confounded or at least overestimated. For example, if older people are more likely to live alone, this would explain why older people have lower scores on this component (rather than actual changes in relationship quality). Similarly, if “time with people” really means “time with housemates”, it may be tautological to say that “People living alone score...lowest for more time with people” (p. 14 in the PDF). Another example of a potentially inaccurate sentence: “people with psychiatric and neurologic conditions were less likely to report increased connectedness during the pandemic” (p. 22). One way to deal with this would be to control for whether participants are living alone, or perhaps exclude such participants from analyses involving this component.

We concur with the reviewer that cohabitation status likely relates to this particular PD-GIS domain. It also is the case that older adults are more likely to live alone (e.g., about 2% of people in their twenties vs about 15% of people in their 70s). People who live alone make up a substantial proportion of participants (~11K) and we agree that they likely are affected quite differently. Nonetheless, when we reran the PD-GIS factor analysis excluding people who live alone, the overall principal component analysis structure remained highly similar. Specifically, the PD-GIS showed 7 significant components (based on permutation analysis) with or without inclusion of those who live alone and the correlation across the component loading matrices is $r=0.95$. Consequently, we have kept the PCA as reported. We note that the analyses downstream from the PCA all already account for the unique contribution of living alone. More specifically, living alone is a discrete binary predictor in the linear models reported in figures 6 and 7. Therefore, this cannot be the basis of the relationship observed with age, for example. We have modified the text referring to this analysis. The bottom right figure in 7 shows the clear relationship of living alone to this component, but also a substantial relationship to other aspects of living arrangements. The subsequent analyses reported (e.g., of psychiatric conditions and personality

traits etc) have the variables from that first linear model factored; therefore, they also account for the contribution to the variance of living alone.

2. I would like to see more context on how the PD-GIS was developed. Who came up with the items? What were the criteria for writing and retaining items? Did you initially generate a larger pool of items, and if so, how did you end up with the final set of items?

The PD-GIS was generated in response to the need for a scale that captured aspects of how people considered their daily lives to have been affected, that is, as opposed to more generic measures of their mental health status. The items were generated by the authors of this article, comprising psychiatrists, psychologists and neuroscientists. At a coarse grain, it was designed to have three main sub-sections. (1) Aspects of positive impact. (2) Aspects of negative impact. (3) Outlook. The exact wording of the scale was refined through multiple iterations by the researchers, who are experienced in developing new scales, with feedback from producers at BBC and BBC media, who are knowledgeable regarding the wording of questions such that people will be comfortable answering them online. Prior to application in the main study, the preliminary scale was deployed in ~1,000 participants as a pilot. On analysing the pilot data, it was observed that there appeared to be multiple dimensions to the PD-GIS latent variable structure, that these were quite independent within the positive and negative domains, and that the item-component loadings were relatively simple and readily interpretable. Therefore, the full-scale study included the scale as originally designed, so no items were removed.

This information has been added to the supplement.

Note that the editor asked whether the removal of the free text analysis is detrimental to the impact of the work. Although I thought that the free text analysis was one of the most intriguing aspects of the previous manuscript, I also felt that the previous manuscript was trying to do too much (at least for a short report). The revised manuscript already contains more than enough data without the free text analysis. I hope to see the free text analysis of the advice topics reported as a separate paper.

Thank you - we agree and look forward to sharing the free text analyses in future work.

However, the authors might consider including the free text analysis of the positive and negative impacts in the supplement to provide further context on the PD-GIS. Doing so would demonstrate the extent to which the PD-GIS comprehensively captured the positive and negative impacts that participants spontaneously generated (e.g., Did the PD-GIS leave out any frequently-mentioned impacts? As mentioned in my previous review, how do the PD-GIS dimensions conceptually map onto the topics identified via LDA?).

We agree that juxtaposing the PD-GIS with the NLP analyses has substantial merit - this is why we included the free text fields. However, publishing these data in the supplement would preclude our intended separate paper (about to be submitted now we know it is not included in the main text here) that analyses the NLP data in detail. We have updated the limitations section to note that the PD-GIS may not capture all domains that people find important in terms of pandemic impact, and that in future work we shall analyse free text to address this issue. Now that we have confirmed that the NLP analysis is not being included in this article, we are able to proceed with submission in a further article, which we already have prepared.

3. Some of the questions in the PD-GIS were not worded as increases or decreases, and therefore cannot be interpreted as "impact" of Covid-19. For example, the two questions about health concerns ("I am concerned about my personal health"; "I am concerned about the health of my loved ones") were not worded as being "more" concerned. In the absence of a comparison against a pre-Covid-19 baseline, differences between groups cannot be interpreted as differences in increased health concerns due to Covid-19 (e.g., "increased health concerns scores higher for people with pre-existing concerns...but were greater still for people with anxiety disorders and OCD", p. 16). Instead, such differences may reflect general differences in health concerns irrespective of the Covid-19 context.

In general questions were worded as increases or decreases - we apologise that this was not sufficiently clear previously. Furthermore, all items were contextualised by 'Please indicate how well the following statements describe the impact of pandemic on you.' The full wording of the items noted above were in fact phrased in terms of the effects of the pandemic. E.g., 'I am MORE concerned about my personal health'. We have made sure full wording of each item is reported correctly in the supplement now - there were two errors, both omission of the word 'now'. Similarly, the questions were framed to ask specifically about the impact of the

pandemic, so they would not reflect general differences in health concerns unrelated to the Covid-19 context. Again - all items were contextualised by 'Please indicate how well the following statements describe the impact of pandemic on you.' And for this subsection also by 'How will things change in the long term?' Furthermore, the full wording of items generally were phrased in terms of the effects of the pandemic. E.g., 'I believe the negative impact on the economy will be short lived'. We have updated the manuscript and supplement to make the above clear.

To ensure accuracy, the authors should rewrite sentences like the above in terms of group differences rather than impact (e.g., "People with pre-existing conditions had higher health concerns compared to people who did not have pre-existing conditions"). The authors should also do more to discuss the limitations of measuring impact using retrospective self-reports (compared to a longitudinal design). This was a concern raised by a previous reviewer that was not sufficiently addressed.

As noted above, the statements were all carefully couched in terms of self-perceived impact of the pandemic. We have reworded interpretation throughout to state 'self-perceived impact' as opposed to 'impact' when referring to our specific analyses. We have noted the complexity of inferring impact from self-perceived impact within the discussion section, but we also highlight the particular value of considering how people perceive themselves to have been affected.

4. In the previous version of the manuscript, the authors identified 11 underlying dimensions. In this version, there are 7 underlying dimensions. Without seeing the author's response to the reviews, it is not clear to me how the analyses changed to produce this rather different summary of the PD-GIS.

We understand that the previous response-to-reviewers was not sent to the reviewers due to an editorial oversight but should now have been forwarded. We use a robust permutation-based approach to estimate N components. This method, adopted in response to your previous comments, is described in the supplement as follows.

'A MATLAB implementation of Horn's Parallel Analysis was applied to estimate the number of significant components from the principal component analysis. PCA. This is a permutation-based approach whereby the true data are permuted and data reduced with PCA many times, producing distributions of variance explained by components at each index for statistical comparison to those observed for the unpermuted data. Estimated with 1000 permutations indicated 7 statistically significant components (greater than 95% of values within the corresponding null distribution) when the PD-GIS data were analysed in this manner. Application of the Kaiser convention would indicate 11 components with eigenvalues >1.'

5. I am wondering why the authors chose to use component scores rather than to create scales based on the highest-loading items for each component. I see two major downsides to using component scores: a) because a PCA is likely to produce different solutions in different populations, other studies that use the PD-GIS are unlikely to have the same weights or even the same number of components, making comparisons between studies difficult, and b) the components cannot be cleanly interpreted in terms of a few conceptually related items (instead, they capture a blend of all of the impact items, just to varying extents). Relatedly, in the Method (p. 25), the authors state "Where appropriate, scales were summarised in the following steps...subscale scores estimated using principal component analysis". Does this mean that all scales were estimated using PCA? For example, it was unclear to me whether PCA was used only to produce a six-factor summary of the Big Five measure, and if scale means were then used (e.g., if "Introverted" only included "Talkative", "reserved", "tends to be quiet"), or whether component scores that included all 18 items were used. The latter approach would be extremely unconventional.

The PD-GIS can be analysed at the component or item level. Here we report both dependent on the question being addressed. Specifically, the item level is analysed for general population trends (figure 4). The component analysis seeks to determine first whether it is the case that peoples' self-perceived impact can be explained on just one axis - i.e., positive vs negative - which is not the case, and then given the statistical evidence for multiple components, to characterise them and see how they vary with other population variables. This data reduction step prior to more detailed analysis is warranted because item level analysis in that detail would be unmanageable, and furthermore there is a robust latent variable structure capturing shared variance across multiple items. More broadly, in all cases, PCA was applied in a data driven manner including all item scores. This is the standard exploratory approach. Including highest loading items only would not be statistically appropriate in our opinion as the aim of factor analysis is to reliably estimate latent variables that underlie variance across multiple items. We concur though that further item level analyses could be conducted and would

have substantial value. We are currently undertaking several more focused analyses for further articles and we will make our data available to others who may wish to focus on other more specific questions. We are also mindful that one of the other reviewers felt this aspect was already given too much space in the paper, and we cannot include all possible analyses in one article as there is insufficient space and the message would be lost.

6. *p. 19 states that self-security and compassion were associated with reduced scores on disrupted lifestyle and health concerns, but Figure 10 seems to show that only self-security is associated with reduced scores (and compassion is increased with increased scores, but there is no indication of significance in the figure). This is mentioned again in the discussion (“positive impact with compassion”, p. 22), but does not seem to be true. That sentence in the discussion also mentioned extraversion being associated with positive impact, but again, extraversion seems to be negligibly associated with the impact dimensions in Figure 10.*

Apologies, the text has been corrected. Non significant P values after FWE correction are now highlighted by underlining, with this noted in the legend.

7. *The Figure 10 caption should note whether all correlations were statistically significant (it is unclear whether “all $p < 0.0001$ ” refers to each of the correlations, or only to the multiple-predictor models). Also, “light green” should be “light grey”.*

Thank you for noting the typo, this has been corrected. We also now indicate non-significant p values post FWE correction via underlining.

8. *It was unclear which specific result this sentence was referring to, and whether this claim can be substantiated: “Our results indicate an important reason for this observation [about clinically vulnerable people reporting a worsening in their mental health] is likely to be isolation and disconnectedness” (p. 21).*

We have removed this point.

9. *The abstract states that these data are “objective”, but these data are “subjective” in the sense that subjective self-reports were used.*

This has been rephrased.

10. *“the impact of the pandemic on mental health is likely to be mediated by a variety of factors” (p. 3) – “mediated” should be “moderated” or “modulated”*

We agree, this has been rephrased as ‘modulated’.

11. *In Figure 3, the colors need a key (which color corresponds to which direction?). Also, the caption states “Younger adults were less depressed whereas older adults were more depressed”, but this misleadingly implies that younger adults had lower depression scores compared to older adults (whereas the finding is about a relative change from the pre-pandemic baseline).*

The colour key was in the bottom right of the figure. We now also note the colour-valence relationship in the legend text. Legend text also has been amended as requested.

12. *In the supplement, Section 7, what are the estimates? Are they standardized regression coefficients? If so, this should be mentioned.*

That is correct. This text has been updated.

Reviewer #2

I think the current version is much improved and reads very well. I do not have any other comment

Thank you for your supportive comments.

Reviewer #3

I congratulate the authors for their thorough and thoughtful revision of the paper. My primary concerns have been addressed.

We are grateful for your supportive comments and helpful suggestions.

I still feel that the paper presents a bit too much information in the main manuscript. More specifically, a large portion of the paper is focused on presenting predictors of the self-perceived reports of pandemic impact. I do not believe that these retrospective analyses deserve as much prime journal space, especially given that the main contributions of the paper are the comparisons between mental health before and after the pandemic.

We consider the analysis of self-perceived impact on daily life to be a valuable and unique complement to more generic measures of wellbeing/mental health that are typically included in pandemic-related papers. We feel the novelty and value of the paper would be considerably diminished if these aspects were deleted; and note that other reviewers were positive about including these elements. Therefore, we respectfully propose that they be retained.

I also wondered whether the authors might want to more prominently present comparisons or trends over time. In particular, what is currently missing is the initial impacts of the pandemic (e.g., in March, early April). In another manuscript that I just reviewed with different large representative data from the UK, well-being suffered most in the first two weeks after the first case of the virus reached the UK. After the initial shock, well-being gradually returned to normal by June. Beyond these specific findings that I cannot cite for obvious reasons, the key question is whether the relatively minor differences in mental health observed in the two epochs in the current paper might be due to an adaptation effect, whereby people recover after an initial drop in mental health.

We agree with the suggestion that more detailed timecourse analyses would be valuable. In fact, we are undertaking analyses at the moment that examine such detailed timecourse data, including intersection with area and lockdown status. However, there is not space for such detailed analysis of timecourses to be presented and discussed in this first article from this large dataset. What we can state confidently, is that the differences we observe in anxiety levels between January and May are sustained throughout the entire month. Therefore, we view it as extremely unlikely that they are brief effects (this was shown in one of our original figures but was removed/replaced on request at the last round of reviews).

Finally, a super minor comment to consider replacing "vs." with "by" in the titles of many of the figures.

We agree and have made this alteration as requested.

Thank you.

Reviewer #4

The authors have done a very nice job with their revisions. The manuscript's presentation and clarity are a huge step up from the previous version, and it is my perspective that they have put in a lot of work into making this a nice, linear manuscript with a clear purpose and goal, and with the analyses being very well-documented and described. In these regards, I think that this is a highly successful revision. The manuscript is now much more well-grounded in several relevant literatures, and I again appreciate the thoughtfulness that went into this revision.

Thank you.

*The only real lingering critique that I would have to offer is still a bit of a hold-over from my initial review: the ultimate presentation of this work feels as though it's a cursory "description" of phenomena that we already expected, anticipated, and have been shown elsewhere in various forms. I will reiterate that there is certainly value in this — however, it leaves me wanting a deeper/more extensive interpretation of the patterns. The correlations themselves are intuitive and gel with what we'd expect, and there aren't any earth-shattering revelations: specific psychodemographic subpopulations who are already at-risk for in any stressful/taxing scenario (short and long term alike) are experiencing those issues in a more pronounced fashion during covid. I would nudge the authors to go further than they are currently going: there is already a sense of what we (i.e., policymakers) *should* be doing with this information. What does this information suggest we should do that*

might not have been suggested by past work? For example, does this information help us to identify subpopulations that might benefit from post-hoc assistance? If so, what forms of assistance? And so on.

We are grateful to the review for these supportive comments. We agree regarding the value of more detailed analyses and note that we will be producing further, more focused, articles based on this unique dataset, and are making it available for others to do so as well. We are already at the word limit and the editor has requested that we do not overextend in terms of policy suggestions. Nonetheless, we have substantially reworked the discussion with this angle in mind and somewhat expanded our thoughts in this respect in the process. This includes the following topics covered in the discussion: the need for greater emphasis on pandemic impact in older people (largely overlooked); the role for green space and importance of making this suitable for vulnerable groups; that people with pre-existing conditions are impacted in different ways (as exemplified by OCD and ADHD, for example); positive impact of furloughing; and the need to consider technology use and compulsivity in terms of pandemic impact, rather than just conventional personality measures.

REVIEWER COMMENTS

Reviewer #1 (Remarks to the Author):

I thank the authors for their thoughtful clarifications and revisions. I am satisfied with these revisions, and commend the authors on an impressive contribution.

Final minor points:

1. The authors state in response to point 11 that they changed the figure 3 caption, but the text still incorrectly reads: “Younger adults were less depressed whereas older adults were more depressed”. This should be edited to make sure that it reflects a group difference in change, rather than a group difference in means.
2. Typo on line 556: “di vs did not”. Also, “looking forwards” (used in a couple of places) should be “looking forward”.

Also just a comment (no action required): The authors note in response to point 5 that “the aim of factor analysis is to reliably estimate latent variables that underlie variance across multiple items”. If this was the goal, the authors should have used EFA or CFA (which are measurement models of latent variables), rather than PCA (which reduces data to a linear combination of variables). In the case of the PD-GIS, PCA is appropriate because most of the components (with the exception of the positive outlook items) seem more formative (i.e., they represent a *summary* of the indicators) than reflective (e.g., it seems unlikely that a “disrupted lifestyle” latent variable exists that causes responses on the items). But for what it’s worth, for personality variables (speaking as a personality psychologist), it’s unconventional to model these as a linear combination of all of the items (including items that were not intended to measure the focal trait). It’s much more standard to model them as observed scale means or latent scores from CFA models, because we assume that items like “talkative” and “quiet” are being caused by an underlying latent variable (extraversion). This may be a disciplinary difference.

Signed: Jessie Sun

Reviewer #3 (Remarks to the Author):

I have no further suggestions. However, I would like to highlight that the authors argue against both of my substantive suggestions from my previous review.

First, I suggested that the authors trim the section on perceived impacts. The authors argued that deleting the section on perceived impacts would diminish the value of the paper. I never suggested that this section be deleted, but just streamlined. Thus, I believe my suggestion was not substantively addressed either in the paper or in the rebuttal.

Second, I suggested that the authors present more detailed timecourse analyses. The authors' response was that there was no space for such detailed analyses. They seem to suggest that these analyses will be saved for a different report. If the perceived impacts section of the paper is not streamlined as I previously suggested, it is, of course, an issue to include additional analyses.

In short, the authors have ignored my suggestions for both additions and reductions to their manuscript in their revisions. I still maintain that my suggestions would make the paper stronger. However, I accept that the authors may reasonably disagree with my suggestions as they have outlined in their rebuttal. I leave it up to the editor to decide whether the authors have made a sufficient case that their paper will be stronger and make a greater contribution without incorporating the suggestions I made in the previous review and briefly outlined here.

Reviewer #4 (Remarks to the Author):

I am satisfied with the authors' responses and approve of publication of this paper.

RE: Dimensions and modulators of behavioural and mental-health change during the Covid-19 pandemic. Ref NCOMMS-20-28403-T

We are grateful for the reviewers encouraging and helpful comments. All points raised have been addressed below and the article has been updated. We feel that it is greatly strengthened and hope that you will now consider it ready for publication.

Adam Hampshire (on behalf of the authors)

Response to Reviewer Comments

Reviewer #1

I thank the authors for their thoughtful clarifications and revisions. I am satisfied with these revisions, and commend the authors on an impressive contribution. Final minor points:

Thank you for your helpful comments.

1. The authors state in response to point 11 that they changed the figure 3 caption, but the text still incorrectly reads: “Younger adults were less depressed whereas older adults were more depressed”. This should be edited to make sure that it reflects a group difference in change, rather than a group difference in means.

- This has now been reworded as follows

“Younger adults showed decreased depression whereas older adults showed increased depression.”

2. Typo on line 556: “di vs did not”. Also, “looking forwards” (used in a couple of places) should be “looking forward”.

- Thank you for spotting these typos, they have now been corrected

*Also just a comment (no action required): The authors note in response to point 5 that “the aim of factor analysis is to reliably estimate latent variables that underlie variance across multiple items”. If this was the goal, the authors should have used EFA or CFA (which are measurement models of latent variables), rather than PCA (which reduces data to a linear combination of variables). In the case of the PD-GIS, PCA is appropriate because most of the components (with the exception of the positive outlook items) seem more formative (i.e., they represent a *summary* of the indicators) than reflective (e.g., it seems unlikely that a “disrupted lifestyle” latent variable exists that causes responses on the items). But for what it’s worth, for personality variables (speaking as a personality psychologist), it’s unconventional to model these as a linear combination of all of the items (including items that were not intended to measure the focal trait). It’s much more*

standard to model them as observed scale means or latent scores from CFA models, because we assume that items like “talkative” and “quiet” are being caused by an underlying latent variable (extraversion). This may be a disciplinary difference.

There has been much debate over the past century regarding application of PCA vs. FA, and the use of exploratory/ data driven vs. confirmatory factor analysis. There is no one correct method for this, some methods are more prevalent than others dependent on exact research topic. In reality, the estimated scores for PCA and FA tend to be very similar, especially when analysing large-scale data. Here, we apply PCA as opposed to FA because this is in alignment across analyses of this article, and with other analyses of this dataset and previous related studies that we have published. It is important to maintain a level of consistency. We also are generally well enough powered to apply exploratory models as opposed to predefined factor or component models, which we believe provides stronger inference as relies on fewer assumptions than confirmatory approaches. In keeping with the general approach in this study, we prefer to report application of models that are not limited/restricted by prior assumptions regarding factor or component structure and note that the PCA models generated have interpretable structures.

Reviewer #3

I have no further suggestions. However, I would like to highlight that the authors argue against both of my substantive suggestions from my previous review.

First, I suggested that the authors trim the section on perceived impacts. The authors argued that deleting the section on perceived impacts would diminish the value of the paper. I never suggested that this section be deleted, but just streamlined. Thus, I believe my suggestion was not substantively addressed either in the paper or in the rebuttal.

We have worked back through this section and sought to reduce wordiness but note that this is difficult to achieve whilst ensuring that readers will be able to understand the results – indeed considerable effort already had been made to be concise.

Second, I suggested that the authors present more detailed timecourse analyses. The authors' response was that there was no space for such detailed analyses. They seem to suggest that these analyses will be saved for a different report. If the perceived impacts section of the paper is not streamlined as I previously suggested, it is, of course, an issue to include additional analyses.

We agree that it is informative to include a plot of the day-by-day timecourses and have generated these for each of the mood self-assessment measures. We have added this within the supplemental materials (supplement 3) due to limits on space (although are happy to move to main text if space is allowed). Specifically, we plot mean and standard deviation scores for each of 30 days post promotion launches in January and May 2020 – that is, pre and mid pandemic. We note that our timecourse analyses are constrained within these timeframes as this is when we have high enough participant numbers to reliably gauge the mood of the nation on an individual day-by-day basis. The results are important because they show consistent differences in those measures through these two epochs, that is, we are not reporting results that are driven by transient differences on a specific heavily sampled day.

In short, the authors have ignored my suggestions for both additions and reductions to their manuscript in their revisions. I still maintain that my suggestions would make the paper stronger. However, I accept that the authors may reasonably disagree with my suggestions as they have outlined in their rebuttal. I leave it up to the editor to decide whether the authors have made a sufficient case that their paper will be stronger and make a greater contribution without incorporating the suggestions I made in the previous review and briefly outlined here.

Thank you for your helpful comments.

Reviewer #4

I am satisfied with the authors' responses and approve of publication of this paper.

Thank you for your positive comments.

REVIEWERS' COMMENTS

Reviewer #3 (Remarks to the Author):

Thank you for your thoughtful and careful revisions.